# Haploinsufficiency of *RREB1* causes a Noonan-like RASopathy via epigenetic reprogramming of RAS-MAPK pathway genes

Oliver A. Kent[1✉], Manipa Saha[1], Etienne Coyaud [1], Helen E. Burston[1], Napoleon Law[1,2], Keith Dadson[3], Sujun Chen[1], Estelle M. Laurent[1], Jonathan St-Germain[1], Ren X. Sun[1], Yoshinori Matsumoto[4], Justin Cowen [1], Aaryn Montgomery-Song[1], Kevin R. Brown [1], Charles Ishak[1], Jose La Rose[1], Daniel D. De Carvalho[1,5], Housheng Hansen He [1,5], Brian Raught[1,5], Filio Billia [3,6], Peter Kannu[7] & Robert Rottapel[1,5,8,9✉]

RAS-MAPK signaling mediates processes critical to normal development including cell proliferation, survival, and differentiation. Germline mutation of RAS-MAPK genes lead to the Noonan-spectrum of syndromes. Here, we present a patient affected by a 6p-interstitial microdeletion with unknown underlying molecular etiology. Examination of 6p-interstitial microdeletion cases reveals shared clinical features consistent with Noonan-spectrum disorders including short stature, facial dysmorphia and cardiovascular abnormalities. We find the RAS-responsive element binding protein-1 (*RREB1*) is the common deleted gene in multiple 6p-interstitial microdeletion cases. *Rreb1* hemizygous mice display orbital hypertelorism and cardiac hypertrophy phenocopying the human syndrome. *Rreb1* haploinsufficiency leads to sensitization of MAPK signaling. Rreb1 recruits Sin3a and Kdm1a to control H3K4 methylation at MAPK pathway gene promoters. Haploinsufficiency of *SIN3A* and mutations in *KDM1A* cause syndromes similar to *RREB1* haploinsufficiency suggesting genetic perturbation of the RREB1-SIN3A-KDM1A complex represents a new category of RASopathy-like syndromes arising through epigenetic reprogramming of MAPK pathway genes.

[1] Princess Margaret Cancer Centre, University Health Network, 101 College Street, Toronto, ON M5G 1×5, Canada. [2] STTARR Innovation Center, University Health Network, Toronto, ON, Canada. [3] Toronto General Research Institute, 100 College Street, Toronto, ON M5G 1L7, Canada. [4] Okayama University Graduate School of Medicine, Dentistry and Pharmaceutical Sciences, Okayama 700-8558, Japan. [5] Department of Medical Biophysics, University of Toronto, Toronto, ON, Canada. [6] Division of Cardiology, University Health Network, Toronto, ON, Canada. [7] Department of Pediatrics, Division of Clinical and Metabolic Genetics, The Hospital for Sick Children, Toronto, ON, Canada. [8] Departments of Medicine and Immunology, University of Toronto, Toronto, ON, Canada. [9] Division of Rheumatology, St. Michael's Hospital, Toronto, ON, Canada. ✉email: kent.uhn@gmail.com; rottapel@gmail.com

The MAPK pathway is an essential signaling cascade that controls cell proliferation, cell survival, motility and differentiation all of which are perturbed in cancer but are also critical to normal development[1]. Activation of RAS, encoded by one of three isoforms HRAS, KRAS, or NRAS, is coupled to the transduction of mitogenic signals from growth factor receptors to multiple downstream effector pathways including RAF-MEK-ERK (MAPK) to regulate transcription[2,3]. The activation of the MAPK cascade involves the sequential phosphorylation of serine/threonine kinases RAF, MEK1/2 and ERK1/2[4]. MAPK pathway regulation and controlled expression of MAPK target genes is essential in mediating diverse physiologic outcomes including cellular transformation, tumorigenesis, developmental disorders, cardiac hypertrophy and heart failure.

Germline gain-of and loss-of-function mutations in genes of the RAS-MAPK pathway lead to the Noonan-spectrum of autosomal dominant disorders, a group of malformation syndromes affecting 1 in 1000 individuals[5–7]. Noonan-spectrum, including Noonan-syndrome (NS), Costello and cardio-facio-cutaneous (CFC) syndromes and others are collectively referred to as RASopathies, display an overlap of clinical features including developmental delay, intellectual disability, craniofacial dysmorphism, and cardiac defects[7–9]. The characteristic craniofacial abnormalities include wide-set eyes, broad forehead, wide nasal base, and downward slanting palpebral fissures[9]. Cardiac hypertrophy and abnormalities are common in the RASopathies[10,11] and NS is a leading cause of congenital heart disease[12]. Germline missense mutations have been identified in PTPN11, KRAS (NS), HRAS (Costello syndrome), and MAP2K1/2 (CFC syndrome)[7]. Although significant efforts have been made in identifying causative genes of NS and NS-like disorders, the underlying genetic causes for up to 20% of NS-like cases are unknown suggesting additional mechanisms of causation[13].

Here, we present a patient with an interstitial 6p25.1p24.3 microdeletion and demonstrate that shared clinical features of previously identified 6p interstitial microdeletion cases[14–18] significantly overlap with the Noonan-spectrum of disorders. The consequences of haploinsufficiency of genes within the 6p25.1p24.3 region remain to be defined. We identify the RAS-effector, RAS responsive element binding protein 1 (RREB1) as the gene commonly lost in 6p interstitial microdeletion cases for which there is genomic data. RREB1 is a large zinc-finger transcription factor (TF) implicated in RAS signaling and cancer[19]. RREB1 regulates transcription by binding to RAS-response elements (RRE) in target promoters downstream of RAS-MAPK pathway activity[19–23]. The clinical relevance of RREB1 expression, the target genes RREB1 controls and its role in regulating the RAS-MAPK pathway in human tissues is poorly understood. The mechanism by which RREB1 regulates transcription remains unknown. We find Rreb1 hemizygous mice display orbital hypertelorism and age dependent cardiac hypertrophy phenocopying the human 6p25.1p24.3 syndrome. Rreb1 haploinsufficiency resembles a RASopathy-like syndrome and leads to sensitization of MAPK signaling in fibroblast and cardiac cells. RREB1 recruits SIN3A and KDM1A to an RRE in target promoters in human and murine cells to control histone H3K4 methylation of MAPK pathway genes.

## Results

### RREB1 haploinsufficiency causes a RASopathy-like malformation syndrome.
We identified an eight-year-old boy who exhibited short stature, mild intellectual disability, developmental delay and widely spaced eyes (Fig. 1a) (complete case report in Supplementary Note 1). The facial appearance and other features were reminiscent of a Noonan-spectrum (NS) disorder. Diagnosis

of NS-like syndrome was explored by whole exome sequencing but no pathogenic variants in 13 known NS genes were identified. A chromosomal array analysis revealed a heterozygous 2.1–2.7 Mb interstitial microdeletion of 6p25.1p24.3 which contained 11 genes (Fig. 1b).

Examination of phenotypes associated with 6p interstitial microdeletion cases referenced in the literature[14–18] and cases reported in the NCBI-ClinVar database with copy number loss involving 6p25.1p24.3 display an overlap of clinical features including intellectual disability, craniofacial dysmorphism, and cardiac abnormalities (Fig. 1c, Supplementary Fig. 1a). Examination of the break points reported for these fourteen 6p interstitial microdeletion cases revealed the shortest region of overlap which contained only the RREB1 gene (Fig. 1d).

We examined RREB1 expression in EBV-transformed lymphoblastoid cells (LCLs) derived from the proband and parents and confirmed reduction of RREB1 mRNA and protein in the proband (p < 0.01 t-test, Fig. 1e, Supplementary Fig. 1b). We validated two additional genes contained within the deleted region, SSR1 and LY86, and found they were similarly decreased in LCLs from the proband compared to parents (Supplementary Fig. 1b, c). LCLs derived from the proband had significantly increased phosphorylated (p)-MEK and p-ERK when stimulated with FBS with increased proliferation compared to LCLs derived from either parent (p < 0.0001 t-test, Fig. 1f, g and Supplementary Fig. 1d, e) consistent with aberrant RAS-MAPK pathway activity associated with a RASopathy-like syndrome.

To model RREB1 haploinsufficiency as the driver of a malformation syndrome, we generated a Rreb1 knockout mouse model. We used CRISPR-Cas9 to disrupt the coding frame of murine Rreb1 in C57BL6 zygotes (Fig. 2a, Supplementary Fig. 2a). Heterozygous (Rreb1+/−) progeny were obtained but following heterozygous crosses no Rreb1−/− mice were obtained. Examination of the uterus from pregnant females revealed multiple implantation sites with no embryonic remains suggesting Rreb1−/− embryos died after implantation prior to E9.5 (Supplementary Fig. 2b)[24]. Thus, homozygous Rreb1 deletion is lethal during early embryogenesis, whereas one copy of the Rreb1 allele is sufficient for postnatal viability. Rreb1+/− mice were slightly smaller with a wider blunted nose and significantly increased intercanthal distance (ICD, p < 0.001 t-test) compared to WT littermates (Fig. 2b–d and Supplementary Fig. 2c, d) similar to the changes in the cranial bones observed in the Noonan syndrome mouse model[25].

Since cardiac hypertrophy and dysfunction are commonly observed in patients with 6p microdeletion and Noonan-like syndromes[7–9,14–18], the hearts from Rreb1+/− and WT littermates were compared. Rreb1 mRNA was expressed at approximately half in Rreb1+/− hearts (Supplementary Fig. 2e). Rreb1+/− mice developed left ventricular hypertrophy at 6 months of age with pronounced cardiac wall thickening (Fig. 2e). The cross-sectional area of cardiomyocytes was more than 2-fold higher (p < 0.001 t-test) in Rreb1+/− mice compared with WT and also present in younger animals (Fig. 2f, g and Supplementary Fig. 2f). The heart weight to tibial length of Rreb1+/− mice revealed significantly smaller normalized heart size (p < 0.01 t-test) when compared to WT littermates at 6 months of age (Supplementary Fig. 2g). Analysis of cardiac performance by echocardiography revealed an age related decreased fractional shortening in Rreb1+/− hearts at 6 months of age (p < 0.001 t-test) but not in younger mice (Fig. 2h, Supplementary Fig. 2h, i). Transcripts for the atrial/brain natriuretic factors (ANP, BNP) and β-myosin heavy chain (β-MHC), canonical markers of the fetal gene program and cardiac hypertrophy were up-regulated in Rreb1+/− hearts at 6 months (Supplementary Fig. 2j). RAS-dependent hypertrophic heart

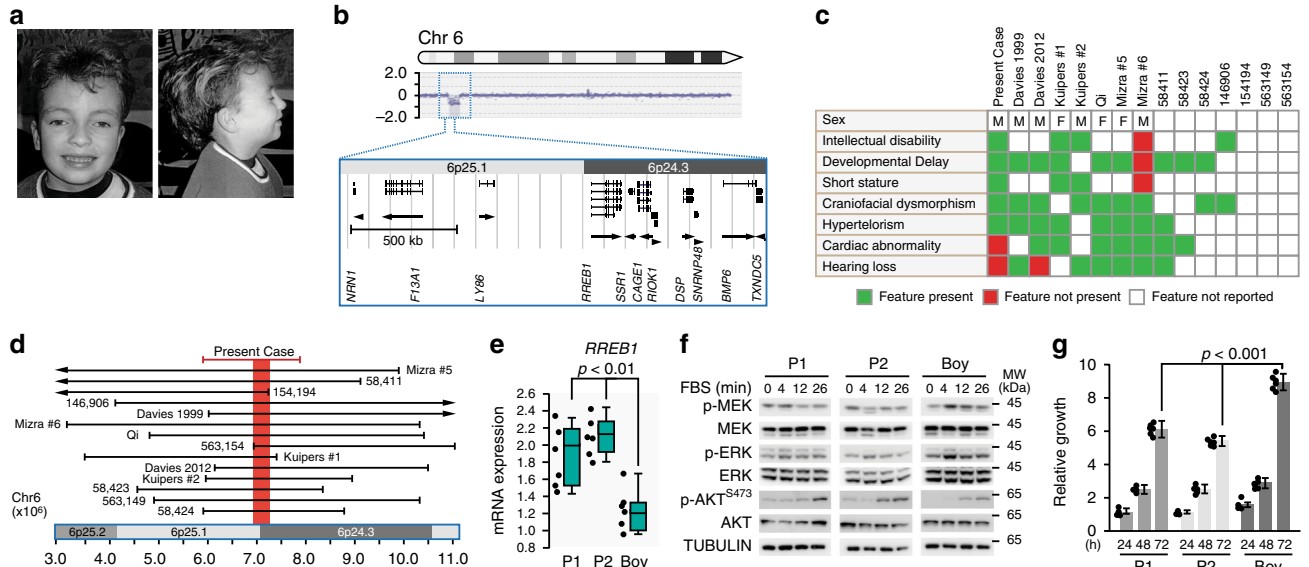

**Fig. 1 RREB1 haploinsufficiency causes a developmental syndrome. a** Photograph of the proband described in the present study. **b** CGC array of the left arm of chromosome 6 identified a heterozygous 2.1–2.7MB microdeletion encompassing 6p25.1p24.3. Expanded region highlights the genes in the microdeletion (UCSC Genome Browser, hg18 assembly). Gene direct is indicated for each gene using an arrow head. **c** Clinical features associated with 6p interstitial deletion cases and **d** overlap of the reported deleted genomic regions involving 6p25.1 and 6p24.3. Case numbers refer to NCBI ClinVar entries. Red line represents the microdeletion in the proband of the present study. **e** mRNA expression of *RREB1* in EBV transformed lymphoblastoid cells (LCLs) derived from the proband (Boy) and parents (P1, P2; *n* = 6). *p* = 0.007 (P1 vs. Boy), *p* = 0.0002 (P2 vs. Boy) two-tailed student's *t*-test, box plots indicate the IQR of the data and the central line shows the median. **f** Western bot analysis of MAPK signaling in LCLs (P1, P2, Boy) following serum deprivation and stimulation with FBS for the indicated times. Experiment was initially a single membrane (provided in Supplementary Fig. 1), separated post exposure for clarity in the figure. Tubulin served as a loading control. Blots are representative of *n* = 6 experiments. **g** Relative growth (arbitrary units) of LCLs (P1, P2, Boy) in culture measured with Alomar Blue. Data average of *n* = 6 independent experiments, *p* = 3.4E-08 (P1 vs. Boy), *p* = 6.3E-10 (P2 vs. Boy) two-tailed student's *t*-test, error bars presented as mean values ± SD. Source data are provided as a Source Data file.

disease in mice is histologically indistinguishable from hypertrophic cardiomyopathy in humans and is a characteristic Noonan-like phenotype associated with gain-of-function MAPK signaling[10,11,26]. Consistent with cardiac hypertrophy observed in *Rreb1*+/− mice and sensitization of MAPK signaling in *RREB1* haploinsufficient cells, increased p-MEK and p-ERK was observed in left ventricle (LV) from *Rreb1*+/− hearts compared to WT (Fig. 2i).

We also examined MAPK signaling as a consequence of *Rreb1* hemizygosity in murine embryonic fibroblasts (MEFs) derived from *Rreb1*+/− E14.5 embryos. *Rreb1*+/− MEFs had reduction in *Rreb1* mRNA as expected (Supplementary Fig. 2k). Similar to the aberrant activation of MAPK signaling observed in cells derived from the proband, *Rreb1*+/− MEFs had increased MAPK activity and proliferation following stimulation with FBS compared to WT (Fig. 2j, k and Supplementary Fig. 2l).

**RREB1 is a negative regulator of RAS-MAPK pathway target genes**. To ascertain the molecular mechanism underlying the negative regulation of the MAPK pathway by *RREB1* haploinsufficiency, we performed RREB1 chromatin immunoprecipitation followed by sequencing (ChIP-seq) in HEK293 cells. Since ChIP grade antibodies are not available for RREB1, we created a tetracycline inducible flag-tagged RREB1 cell line for ChIP experiments. Treatment of cells with tetracycline activated RREB1 mRNA expression 10-fold over untreated cells with RREB1 protein over-expression detected at physiologic levels (Supplementary Fig. 3a, b). ChIP-seq revealed 1239 unique broad peaks and 7488 sharp peaks associated with 3726 genes. RREB1 binding was enriched primarily at promoters centered on the transcription start site (TSS) followed by intronic and distal

regions (Fig. 3a, Supplementary Fig. 3c). KEGG analysis revealed RREB1 target genes enriched in metabolic pathways, MAPK signaling and actin cytoskeleton (Fig. 3b).

To support the ChIP-seq results, we performed RNA-seq in *RREB1* knockdown cells. HEK293 cells expressing a short-hairpin RNA (sh1) targeting *RREB1* had 80% reduction in *RREB1* mRNA and 50% reduction in RREB1 protein compared to control (Supplementary Fig. 3d, e). By RNA-seq analysis, 1457 mRNAs had greater than 2-fold differential expression changes and 87% were upregulated with *RREB1* knockdown. KEGG analysis of RREB1–transcriptome confirmed enriched MAPK pathway target genes which included the small GTPase *HRAS*, multiple MAPKs including *MAP2K2* (MEK2), signaling molecules (*AKT1, DUSP7, FGFR4*) and transcription factors (*JUN, JUND, MYC*) (Fig. 3c, Supplementary Fig. 3f). RREB1 ChIP-seq corroborated regulation of MAPK pathway genes and revealed RREB1 occupancy proximal to a RAS-response element (RRE) on the promoters of the *FGFR4, HRAS,* and *MAP2K2* gene *loci* (Fig. 3d).

RREB1 binding was confirmed on the *JUN, MYC, FGFR4, HRAS* and *MAP2K2* promoters by ChIP (Fig. 3e). RREB1 binding was not detected on the *JUND* or *AKT1* promoters suggesting dysregulation of these genes was an indirect consequence of *RREB1* knockdown (Supplementary Fig. 3g). Previously identified RREB1 targets miR-143/145 (*MIR143/145*) and *ARHGEF2* were used as positive controls for ChIP and corroborated previous results (Fig. 3e, Supplementary Fig. 3g)[23,27].

To demonstrate the direct regulation of target genes by RREB1, we cloned proximal promoters of *FGFR4, HRAS* and *MAP2K2* into the pGL3-luciferase reporter and conducted promoter activation assays in cells with *RREB1* knockdown (Fig. 3f). Promoter activity of all three genes was higher in HEK293 cells

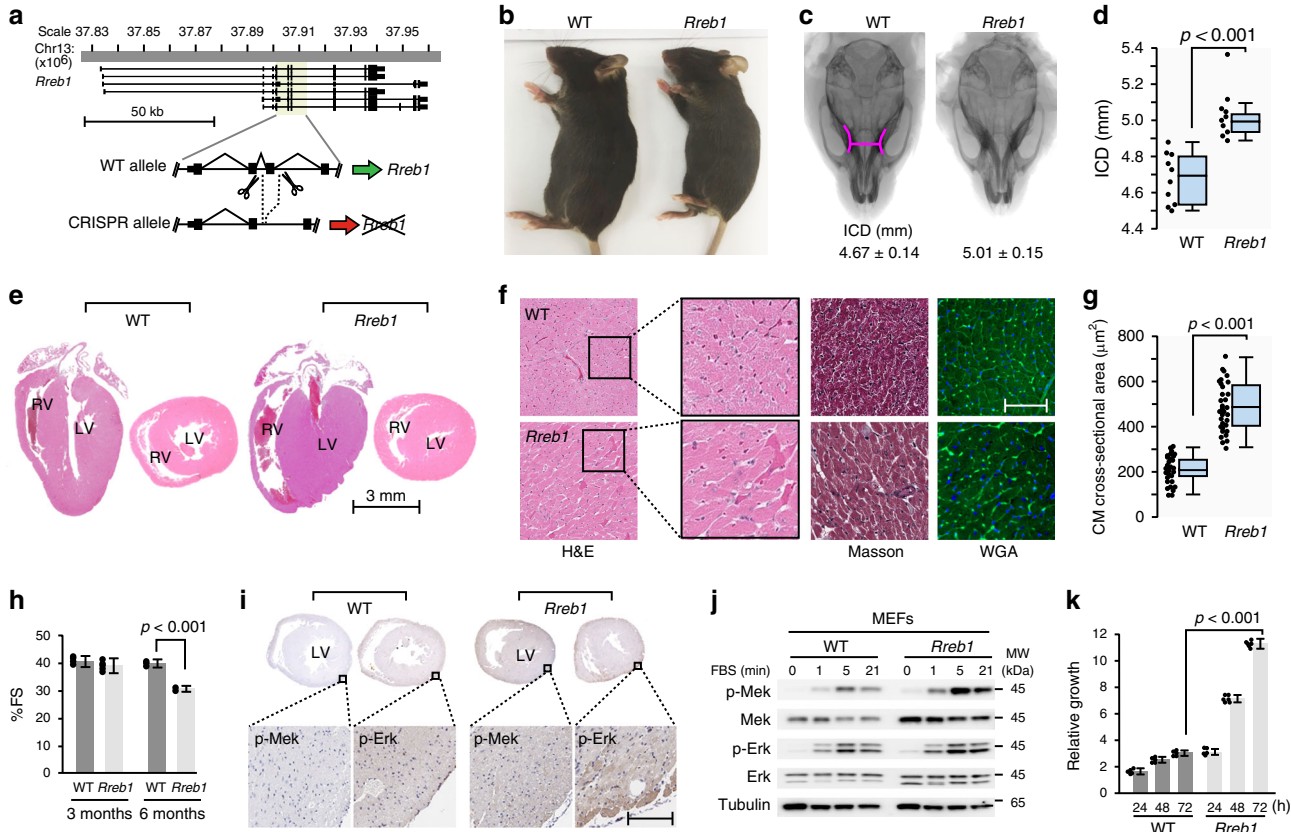

**Fig. 2 *Rreb1* haploinsufficiency in mice resembles a Noonan-like RASopathy syndrome. a** Genomic region around murine *Rreb1* (UCSC Genome Browser, GRCm38/mm10 assembly). CRISPR-Cas9 guide sequences (scissors) flanked an exon common to all splice variants. Cas9 editing results in a premature stop codon which disrupts the modified allele. **b** Gross appearance of male wild type (WT) and *Rreb1+/−* (*Rreb1*) mice. **c** Morphometric x-ray analysis of male WT and *Rreb1+/−* skulls. Pink contour lines mark the edges of the eye sockets used for measurement of intercanthal distance (ICD). **d** Box plot analysis of ICD measurements taken from WT and *Rreb1+/−* skulls (n = 10). Box plots indicate the IQR of the data and the central line shows the median. $p = 3.6\text{E-}05$ two-tailed student's *t*-test. **e** H&E staining of longitudinal and cross-sectional cardiac sections from 6 month WT and *Rreb1+/−* hearts. Heart chambers labeled: LV, left ventricle; RV, right ventricle. Scale bar 3 millimeter (mm). Images are representative of n = 3 hearts examined. **f** H&E, Masson, and wheat germ agglutinin (WGA) staining of LV cardiac sections from 6 month WT and *Rreb1+/−* hearts. Scale bar 100 µm. n = 3 hearts examined. **g** Box plot analysis of cardiomyocyte (CM) cross-sectional area (12 CM measured per heart, n = 3 hearts). Quantification was performed using Aperio ImageScope software. Box plots indicate the IQR of the data and the central line shows the median, $p = 5.6\text{E-}22$ two-tailed student's *t*-test. **h** Fractional shortening (FS%) determined by echocardiography on 3 and 6 month WT and *Rreb1+/−* mice (n = 10). Error bars presented as mean values ± SD, $p = 6.6\text{E-}10$ two-tailed student's *t*-test. **i** IHC analysis of p-Mek and p-Erk in LV cardiac sections from WT and *Rreb1+/−* hearts. Scale bar 100 µm. n = 3 hearts examined. **j** Western blot analysis of MAPK signaling in WT and *Rreb1+/−* MEFs following serum deprivation and stimulation with FBS for the indicated times. Tubulin served as a loading control. Blots are representative of n = 6 experiments. **k** Relative growth of WT and *Rreb1+/−* MEFs in culture measured with Alomar Blue. Data average of n = 6 independent experiments, error bars presented as mean values ± SD, $p = 2.7\text{E-}15$ two-tailed student's *t*-test. Source data are provided as a Source Data file.

expressing sh1-*RREB1* or a second shRNA targeting *RREB1* (sh2) compared to sh-control demonstrating that negative regulation of promoter activity was relieved by *RREB1* knockdown.

MAPK genes *JUN*, *MYC*, *FGFR4*, *HRAS* and *MAP2K2* identified by RNA-seq were upregulated at both the mRNA and protein levels in HEK293 cells expressing sh1 or sh2 targeting *RREB1* compared to control (Supplementary Fig. 3h, i). Consistent with a mechanism of RREB1 repression of target genes, HEK293 cells with the tetracycline inducible flag-tagged RREB1 had repressed levels of *FGFR4*, *HRAS* and *MAP2K2* mRNAs when cells were treated with tetracycline to induce *RREB1* expression (Fig. 3g). Importantly, *HRAS* and *MAP2K2* transcript and protein levels were elevated in LCLs derived from the proband compared to either parent consistent with the conclusion that these are RREB1 dysregulated genes in the patient (Supplementary Fig. 3j, k). *FGFR4* mRNA was not detectable in the LCL cells. Increased

expression of *Fgfr4*, *Hras* and *Map2k2* mRNAs was observed in *Rreb1* deficient fibroblasts and cardiac cells (Fig. 3h, i). Luciferase expression driven by human *FGFR4*, *HRAS* and *MAP2K2* promoters was increased in *Rreb+/−* MEFs relative to WT demonstrating transcriptional repression mediated by *Rreb1* in murine cells (Supplementary Fig. 3l). Endogenous examination of transcripts revealed *Hras* upregulated in *Rreb1+/−* hearts from 3 month old animals which remained high in hearts examined at 6 months while *Fgfr4* and *Map2k2* expression increased in *Rreb1+/−* hearts as animals aged (Fig. 3i, Supplementary Fig. 3m). The proteins encoded by *Fgfr4*, *Hras* and *Map2k2* (Mek2) were similarly upregulated in *Rreb1+/−* hearts (Fig. 3j). Fgfr4 expression was pronounced in LV sections of 6 month *Rreb1+/−* hearts in comparison to WT (Supplementary Fig. 3n). These data provide direct evidence that RREB1 negatively regulates the transcription of key MAPK components and identifies *Fgfr4*, *Hras*

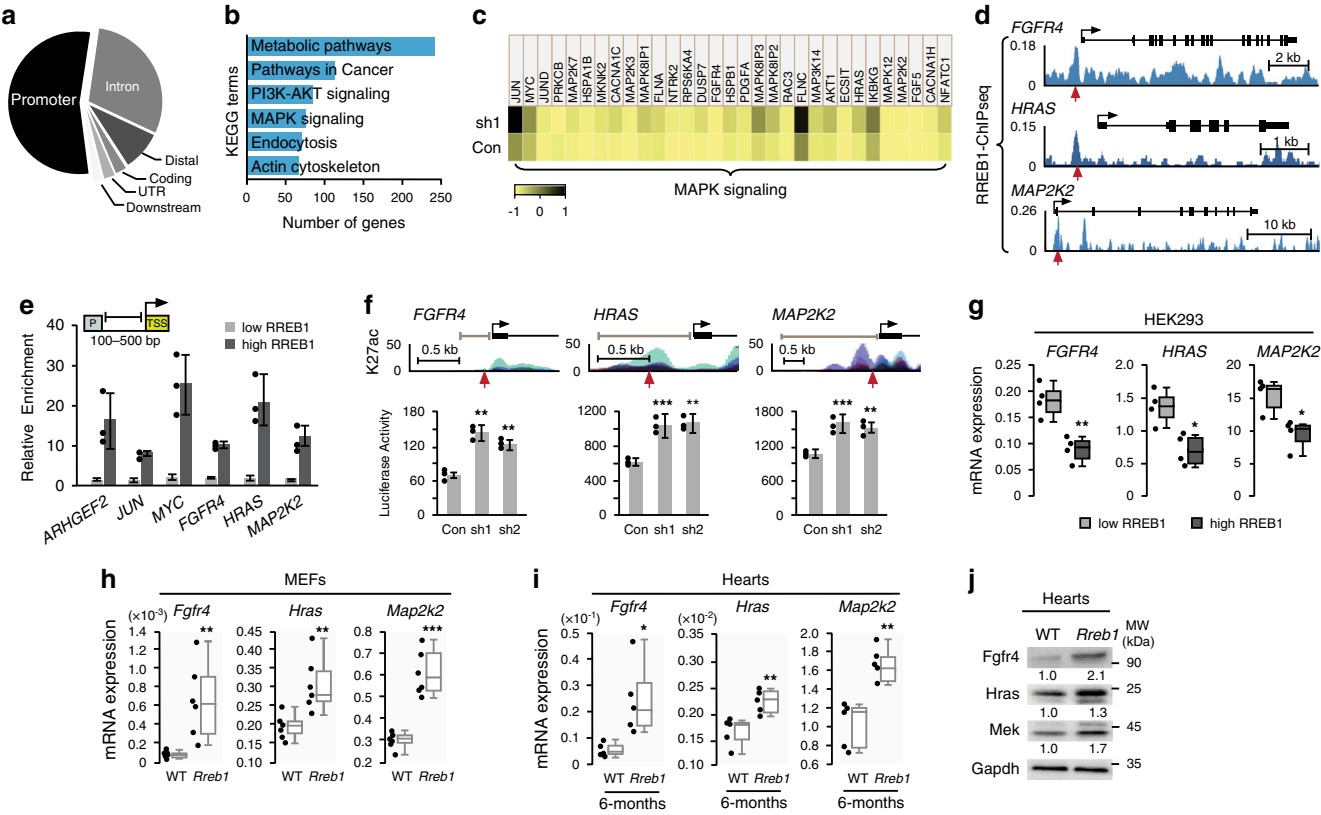

**Fig. 3 RREB1 transcriptionally regulates MAPK pathway target genes. a** Genomic distribution of RREB1 ChIP-seq peaks. Distal (intergenic), UTR (5′ and 3′-UTR), downstream (next to annotated genes). **b** KEGG terms associated with RREB1 target genes identified by ChIP-seq. **c** Heat map view of KEGG defined MAPK pathway genes differentially expressed by RNA-seq analysis in HEK293 cells expressing control shRNA (Con) versus shRNA targeting *RREB1* (sh1). Color scale ranges from yellow lowest expression to black highest expression. **d** Gene track view of RREB1-ChIP-seq peaks at the indicated *loci*. Gene bodies are shown above the track sets. Black arrows of the gene bodies mark the transcription start site. Red arrows point to the RAS response element. **e** RREB1 ChIP-PCR in HEK293 cells with a tet-inducible RREB1 transgene treated with DMSO (low RREB1) or tetracycline to induce RREB1 expression (high RREB1). Amplicons for the indicated genes were queried around promoters (P) designed 100–500 bp from the TSS (inset). For this and subsequent ChIP experiments, enrichment is relative to IgG ($n = 3$). Error bars presented as mean values ± SD. **f** Upper panels depict gene track view of H3K27 acetylation (ENCODE Regulation, Layered H3K27ac track) proximal to the indicated promoters. Red arrows mark the putative RRE. Bar graphs show normalized luciferase activity (arbitrary units) driven from pGL3-reporter constructs with the indicated promoters (gray bars in the upper panels) in HEK293 cells expressing control shRNA (Con) or shRNA (sh1 or sh2) targeting *RREB1* ($n = 3$; **$p < 0.001$, ***$p < 0.0001$ two-tailed student's *t*-test, error bars presented as mean values ± SD). **g** RT-QPCR analysis of the indicated genes in HEK293 cells with tet-inducible RREB1 transgene described in panel **d** ($n = 4$, *$p < 0.01$, **$p < 0.001$ two-tailed student's *t*-test). **h** Expression of the indicated mRNAs in WT and *Rreb +/−* MEFs ($n = 6$) and **i** expression in WT and *Rreb1 +/−* hearts from 6 month old mice ($n = 5$) (*$p < 0.01$, **$p < 0.001$, ***$p < 0.0001$ two-tailed student's *t*-test). Box plots (panels **g,h,i**) indicate the IQR of the data and the central line shows the median. **j** Analysis of the indicated proteins in WT and *Rreb1 +/−* heart lysates derived from 6 month old mice. ($n = 5$). Gapdh served as a loading control. Source data are provided as a Source Data file.

and *Map2k2* as specific genes upregulated in the hearts of *Rreb1+/−* mice.

**RREB1 negatively regulates FGF-HRAS-MAPK signaling.** Since MAPK pathway genes were dysregulated in the RREB1-transcriptome, we hypothesized *RREB1* knockdown sensitized cells to RAS-MAPK signaling. In shRNA-*RREB1* cells, stimulation with FBS increased p-MEK and p-ERK compared to sh-control cells (Fig. 4a, Supplementary Fig. 4a). No change in AKT signaling was observed even though AKT levels were visibly higher with *RREB1* knockdown. *RREB1* knockdown cells proliferated significantly faster ($p < 0.001$ *t*-test) and formed larger colonies in soft agar consistent with elevated MAPK signaling (Fig. 4b, c). Multiple genes are commonly lost in the 6p25.1p24.3 microdeletion cases; therefore we wanted to demonstrate that decreased expression of *RREB1* rather than decreased expression of neighboring genes caused sensitization of MAPK signaling. We examined the expression of the nearest neighbor genes to *RREB1* in HEK293

cells and found they only express *SSR1* in addition to *RREB1* (Supplementary Fig. 4b). Targeting *SSR1* using two independent shRNAs had no effect on p-MEK and p-ERK compared to sh-control cells (Supplementary Fig. 4c, d) confirming a *RREB1* specific role for MAPK pathway regulation.

Next, we examined MAPK pathway activation in response to specific mitogens or cytokines in *RREB1* knockdown cells (Fig. 4d). Treatment with EGF which activates the epidermal growth factor receptor (EGFR), FGF1 or FGF2 which activates FGF receptors (1–4) increased p-ERK in sh1-*RREB1* cells compared to control. FGF16, a ligand specific for FGFR4, strongly activated p-ERK in sh1-*RREB1* cells greater than 2-times compared to control cells. In distinction, FGF8 treatment, which specifically activates FGFR3, had a modest effect on sh1-*RREB1* cells. HGF (hepatocyte growth factor) which activates the receptor tyrosine kinases (RTK) c-MET also increased p-ERK in sh1-*RREB1* cells compared to control. Treatment with TNF-α, a minimal ERK activator or TGF-β, which does not activate ERK, had no enhancing effect in sh-*RREB1* cells. Therefore, *RREB1*

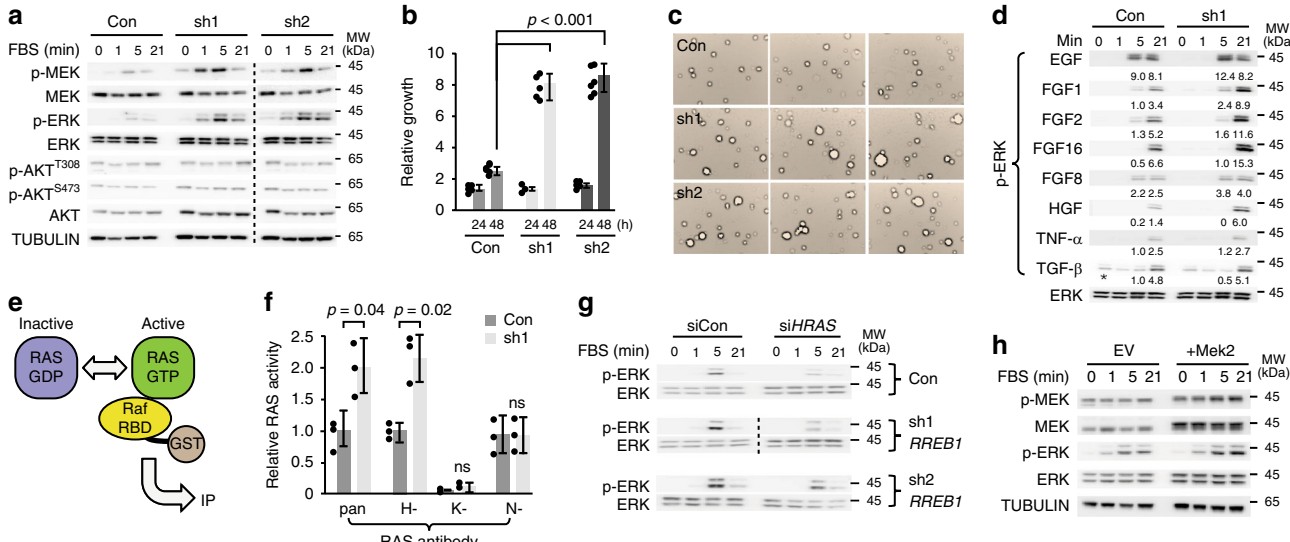

**Fig. 4 _RREB1_ loss leads to increased sensitization of RAS-MAPK signaling. a** Examination of MAPK and AKT signaling in HEK293 cells expressing control shRNA (Con) or shRNA (sh1 or sh2) targeting _RREB1_ following serum deprivation and stimulation with FBS at the indicated times. Dotted line separates different membranes ($n = 6$). Tubulin served as a loading control. **b** Relative cell growth in culture measured by Alomar Blue ($n = 6$, $p = 9.5E-07$ (Con vs. sh1), $p = 5.4E-06$ (Con vs. sh2), value two-tailed student's $t$-test, error bars presented as mean values ± SD) and **c** cell growth in soft agar ($n = 3$) of HEK293 cells expressing sh-control (Con) or shRNA (sh1 or sh2) targeting _RREB1_. **d** Western blot analysis of p-ERK activation in HEK293 cells expressing control sh-control (Con) or sh1 targeting _RREB1_ following stimulation with the indicated mitogens or cytokines. Total ERK blot is representative for all panels. Quantification (5 and 21 min) time points indicated, normalized to TGF-β treated zero time point (asterisk) ($n = 2$). **e** Schematic of the RAS activation assay. RAS cycles between GDP (inactive) and GTP (active) bound forms. Active RAS can be trapped by affinity binding to the Raf-Ras binding domain (RBD) fused to glutathione S-transferase (GST)-tag. The GST tag is used for immunoprecipitation (IP) of activated RAS. **f** Bar graph summary of RAS activation in HEK293 cells expressing sh-control (Con) or sh1 targeting _RREB1_ (sh1). Activated RAS (RAS-GTP) was calculated using the ratio of RAS-GTP over total RAS quantified from western blot analysis following 3 min stimulation with FBS ($n = 3$, n.s. = not significant, $p$-value two-tailed student's $t$-test, error bars presented as mean values ± SD). Pan-RAS directed antibody detects all RAS isoforms, specific RAS antibodies are indicated (H-, K-, N-). **g** Analysis of p-ERK following stimulation with FBS for the indicated time in HEK293 cells stably expressing sh-control (Con) or shRNA (sh1, sh2) targeting _RREB1_, 72 h post transfection with control (siCon) or siRNA targeting _HRAS_ (si_HRAS_) ($n = 3$). **h** Analysis of MAPK signaling in HEK293 cells expressing empty vector control (EV) or Myc-tagged Mek2 (+Mek2) following stimulation with FBS at the indicated times ($n = 4$). Tubulin served as a loading control. Source data are provided as a Source Data file.

negatively regulates MAPK signaling specifically downstream of EGFR, FGFR4 and MET-RTKs.

Since FGFR4 and RTKs activate the canonical RAS-MAPK pathway, we examined the RAS isoform principally activated in _RREB1_ deficient cells. We affinity-purified RAS family members bound to a Raf derived Ras-binding domain fused to GST (Fig. 4e) and observed elevated total RAS-GTP in sh1-_RREB1_ cells compared to control (Supplementary Fig. 4e). Using isoform-specific RAS antibodies the elevated RAS activity was attributed to increased activation of HRAS ($p < 0.02$, Fig. 4f, Supplementary Fig. 4e). This result is consistent with increased gene dosage of _HRAS_ mRNA and protein levels resulting in increased HRAS activation. Therefore, we tested the importance of _HRAS_ expression in sensitizing MAPK pathway activation. Using cell lines expressing shRNA targeting _RREB1_ or control, we depleted _HRAS_ with siRNA to transiently block HRAS signaling and observed decreased p-ERK in sh-_RREB1_ cells (Fig. 4g, Supplementary Fig. 4f). Similarly, we tested whether increased _MAP2K2_ expression was sufficient to increase MAPK signaling. HEK293 cells transfected with a _MAP2K2_-myc tagged transgene had 2-fold more MEK2 (encoded by the _MAP2K2_ gene) and increased p-ERK when stimulated with FBS compared to cells expressing the empty-vector (Fig. 4h, Supplementary Fig. 4g, h). These data show that transcriptional dysregulation of _FGFR4_, _HRAS_ and _MAP2K2_ as a result of _RREB1_ haploinsufficiency is both necessary and sufficient to functionally activate MAPK pathway signaling in different cell types.

**RREB1 interacts with epigenetic chromatin regulation machinery.** Since RREB1 lacks an intrinsic transcription activation domain, the mechanism of RREB1 transcriptional regulation is likely to occur through recruitment of other factors[21]. To understand how RREB1 mediates transcriptional regulation, we conducted RREB1-BioID in the HEK293 model system[28–30]. We identified 241 high-confidence RREB1-interacting proteins using this methodology (1% FDR). The RREB1-interactome comprised sequence specific TFs, transcriptional repressors/activators, and chromatin remodeling proteins (Fig. 5a, Supplementary Fig. 5a). All known members of the CtBP1 complex[31] were detected including 7 high confidence interactors. The RREB1-interactome was composed of transcriptional and chromatin regulators and GO-molecular function identified histone demethylases, regulators of H3K4 methylation and proteins involved in heart development (Supplementary Fig. 5a–c). We validated the interaction between RREB1 and several components using bimolecular fluorescence complementation (BiFC)[32]. Consistent with BioID, intense nuclear fluorescence was observed when cells expressed VC-RREB1 and VN-tagged SIN3A, KDM1A or HDAC1 but not the empty-VN vector which confirmed direct interaction with RREB1 (Supplementary Fig. 5d).

Based on RREB1-BioID, we hypothesized that RREB1 is a regulator of epigenetic repression affecting histone H3 lysine-4 di-and tri-methylation (H3K4me2, K4me3) and lysine-9 acetylation (K9ac) typically found at active promoters[33]. Consistent with this hypothesis, global histone analysis revealed increased

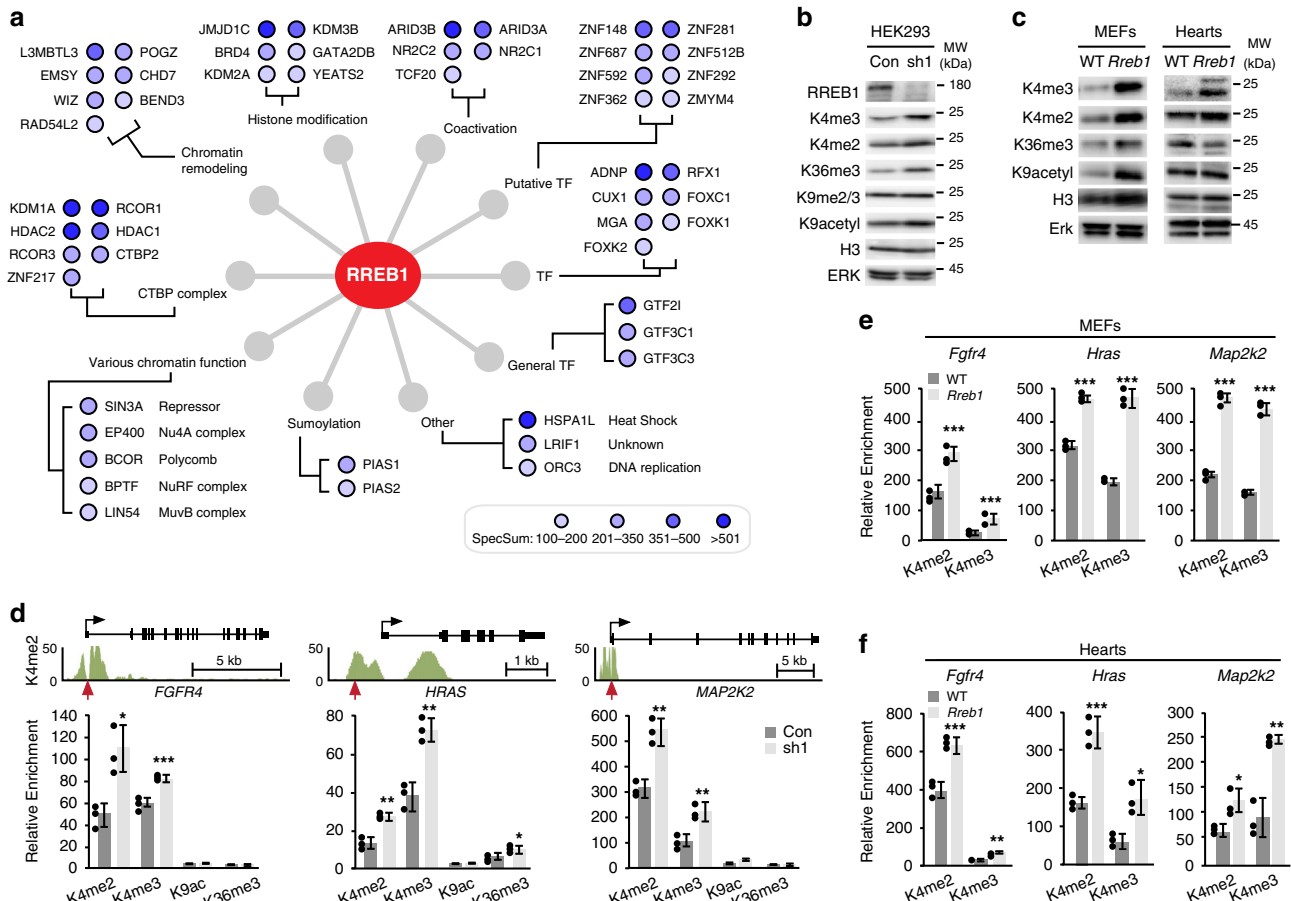

**Fig. 5 RREB1-BioID reveals a mechanism of epigenetic histone regulation. a** RREB1 interactors identified by BioID and classified according to information compiled at NCBI-Gene, Gene Cards, and literature searches. Dot color intensity represents spectral sum counts (SpecSum) as indicated in the scale. **b** Western blot analysis of histone modifications in HEK293 cells expressing control shRNA (Con) or sh1-*RREB1* (sh1)and **c** histone modifications in WT and *Rreb1*+/− MEFs and 6 month LV total lysates (Hearts). H3 and ERK are loading controls. All blots representative of *n* = 3 independent experiments. **d** ChIP analysis of H3K4me2 modification at the indicated promoters. Upper panels show the gene track view of H3K4me2 ChIP-seq (ENCODE, Broad Institute, K562 cells) at the *FGFR4*, *HRAS* and *MAP2K2* loci. Red arrows point to the RRE. Lower graphs show ChIP-PCR of the indicated histone marks at promoters from HEK293 cells expressing sh-control (Con) or sh1-*RREB1* (sh1, *n* = 3). **e** ChIP analysis of histone modifications at the indicated promoters in WT and *Rreb1*+/− MEFs and **(f)** WT and *Rreb1*+/− 6 month hearts (*n* = 3; *\*p* < 0.01, \*\**p* < 0.001, \*\*\**p* < 0.0001 two-tailed student's *t*-test). For panels **d**, **e**, **f** error bars presented as mean values ± SD. Source data are provided as a Source Data file.

H3K4me3, K4me2, K36me3 and K9ac in lysates derived from sh1-*RREB1* and *Rreb1*+/− MEFs compared to control cells (Fig. 5b, c). Increased H3K4me2 and K4me3 were observed in lysates from *Rreb1*+/− hearts compared to WT (Fig. 5c).

We conjectured that *RREB1* loss is associated with increased H3K4me2/3 marks on *FGFR4*, *HRAS*, and *MAP2K2* promoters. We examined H3K4 modification at these target *loci* and found increased K4me2 and K4me3 at the *FGFR4*, *HRAS*, and *MAP2K2* promoters in cells expressing sh1-*RREB1* compared to control (Fig. 5d). Consistent with ChIP, H3K4me2 proximal to an RRE is prominent at these promoters mapped in A562 cells by ENCODE (Fig. 5d). H3K4me2 and K4me3 were enriched on the murine *Fgfr4*, *Hras* and *Map2k2* promoters in *Rreb*+/− MEFs compared to WT (Fig. 5e). H3K4me3 binding is found at the TSS of *Fgfr4*, *Hras* and *Map2k2* promoters near a putative RRE in 8-week old murine hearts (Supplementary Fig. 5e) demonstrating these promoters are active during heart development. We found increased H3K4me2 and K4me3 at all three promoters in *Rreb1*+/− hearts compared to WT (Fig. 5f). These data substantiate that loss of *RREB1* results in epigenetic remodeling associated with transcriptional activity at MAPK pathway genes.

**RREB1 recruits SIN3A and KDM1A to promoters**. We identified a novel interaction between RREB1 and the transcriptional repressor SIN3A (switch-insensitive 3 family member A) with RREB1-BioID. *SIN3A* haploinsufficiency has been described displaying intellectual disability, developmental delay, facial dysmorphia and short stature[34] features similar to the phenotypes of 6p interstitial microdeletion suggesting RREB1 and SIN3A may be part of a common genetic pathway. GO-molecular function analysis of the RREB1-interactome revealed a potential transcriptional repressor complex that included SIN3A, histone demethylases and other proteins (Supplementary Fig. 6a). SIN3A functions as a scaffold with no DNA binding domain or enzymatic activity[35]. We hypothesized that the DNA binding function of RREB1 recruits SIN3A and associated proteins to target promoters to regulate histone modification. Supporting this idea, greater than 70% of the RREB1 ChIP-seq peaks overlapped with SIN3A ChIP-seq and KEGG analysis of the overlap revealed a MAPK signaling signature (Supplementary Fig. 6b).

To define components in a RREB1-SIN3A complex, we precipitated a binary complex that contained VC-RREB1 and VN-SIN3A with an antibody that recognizes the fully complemented Venus (VC + VN) for analysis by mass spectrometry

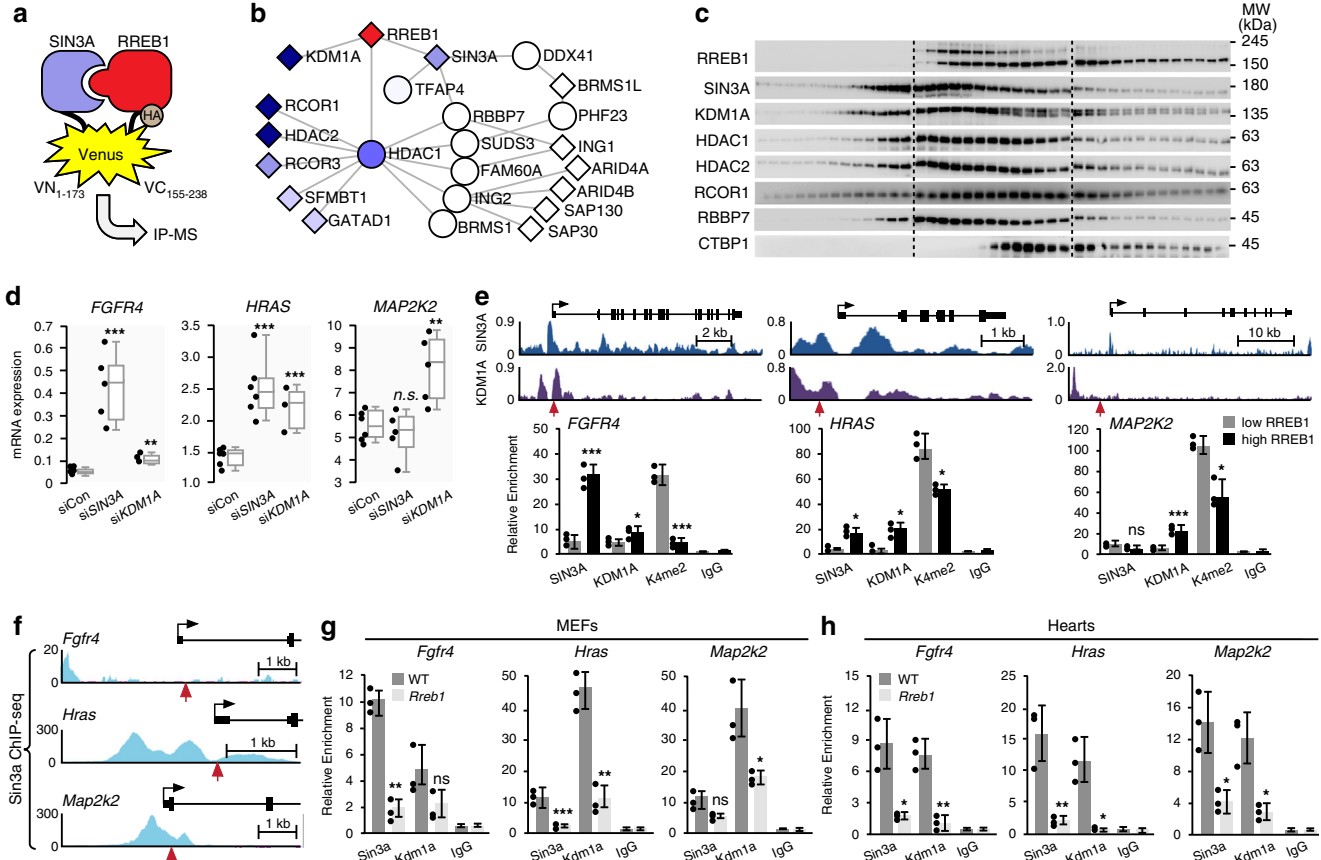

**Fig. 6 RREB1 recruits a SIN3A-KDM1A silencing complex to target promoters. a** Schematic of the protocol for isolation of the RREB1-SIN3A binary complex. RREB1-VC and SIN3-VN interaction reconstitute Venus (VC + VN) as the bait for IP-MS. **b** BioPlex2.0 analysis of the RREB1-SIN3A complex components identified by MS. Bait proteins are represented by a circle and prey proteins with a square. Proteins also identified in the RREB1-BioID are color coded based on SpecSum counts described previously. **c** Western blot analysis of protein fractions isolated by glycerol gradient (60–10%) ultracentrifugation. Blots were probed with the indicated antibodies for endogenously expressed RREB1 complex components. Dashed lines indicate different membranes ($n = 3$). **d** Expression of the indicated mRNAs in HEK293 cells expressing control siRNA (siCon) or siRNAs targeting *SIN3A* or *KDM1A* ($n = 6$). Box plots indicate the IQR of the data and the central line shows the median. **e** ChIP analysis of RREB1-SIN3A complex binding at the indicated promoter regions. Upper panels show the gene track view of SIN3A ChIP-seq (ENCODE, CistromeDB 63655, MCF-7 cells) and KDM1A ChIP-seq (ENCODE, CistromeDB 64823, K562 cells) at the indicated loci. Gene bodies are shown above the track sets for orientation. Red arrows point to the RRE. Lower panels show SIN3A, KDM1A and H3K4me2 ChIP-qPCR in HEK293 cells with a tet-inducible RREB1 transgene treated with DMSO (low RREB1) or tetracycline (high RREB1) ($n = 3$, error bars presented as mean values ± SD). **f** Gene track view of Sin3a ChIP-seq (ENCODE, Stanford/Yale TFBS, MEL cells) at the murine *Fgfr4*, *Hras*, and *Map2k2* proximal promoters. Red arrows point to the putative murine Rreb1 binding elements. **g** ChIP analysis of Sin3a and Kdm1a at the indicated promoters in WT and *Rreb1 +/−* MEFs and **h** WT and *Rreb1 +/−* 6 month hearts ($n = 3$; *$p < 0.01$, **$p < 0.001$, ***$p < 0.0001$ two-tailed student's *t*-test, error bars presented as mean values ± SD). Source data are provided as a Source Data file.

(IP-MS) (Fig. 6a, Supplementary Fig. 6c). We used BioPlex (biophysical interactions of ORFeome-based complexes) Network 2.0[36] to analyze the IP-MS dataset against validated protein-protein interactions. Using this approach, we discovered a minimal RREB1-SIN3A complex (Fig. 6b) that contained SIN3A and associated factors[37] and known components of a CoREST complex[38]. In addition, the RREB1-SIN3A complex was coupled to proteins involved in RNA splicing, RNA end formation, and negative regulation of transcription thus connecting downstream RNA processes to the activity of the RREB1-SIN3A complex (Supplementary Fig. 6d).

To validate the existence of an endogenous RREB1-SIN3A complex, we performed glycerol gradient ultracentrifugation of lysates derived from HEK293 cells followed by western blot analysis of isolated fractions. We observed that all three isoforms of endogenous RREB1 (three human isoforms MW 188, 181 and 159 KDa) co-eluted in the same fractions with SIN3A, RBBP7, HDAC1/2, KDM1A and RCOR1 consistent

with a common macromolecular complex (Fig. 6c). Notably, RREB1-SIN3A eluted earlier from CTBP1 suggesting a distinct complex from RREB1-CTBP1. CTBP1 appeared to be predominantly associated with the 159 KDa isoform of RREB1 (Fig. 6c).

Recently, KDM1A has been shown to be part of a SIN3A-HDAC complex and is functionally associated with the same promoters[39]. In a reciprocal BioID screen, we identified RREB1 as a high confidence interactor using KDM1A as the bait (Supplementary Fig. 6e). We hypothesized that RREB1 target gene regulation depended on the recruitment of SIN3A-KDM1A to promoters. We knocked down *SIN3A* or *KDM1A* in HEK293 cells and observed increased *FGFR4* and *HRAS* mRNA in cells expressing siRNA targeting *SIN3A* and increased *HRAS* and *MAP2K2* mRNA in cells expressing siRNA targeting *KDM1A* (Fig. 6d, Supplementary Fig. 6f). Dysregulated expression of these genes in either *SIN3A* and/or *KDM1A* knockdown cells recapitulated sensitization of MAPK pathway signaling we

observed in *RREB1* haploinsufficient cells providing evidence that modification of H3K4 methylation is sufficient to amplify MAPK pathway signaling (Supplementary Fig. 6f–h).

To establish that recruitment of SIN3A-KDM1A to promoters was dependent on RREB1, we performed SIN3A and KDM1A ChIP in HEK293 cells using Tet inducible RREB1. SIN3A and KDM1A were found in proximity to an RRE in the *FGFR4*, *HRAS* and *MAP2K2* promoters mapped in MCF-7 cells (SIN3A) or K562 cells (KDM1A) by ENCODE (Fig. 6e). In cells with high RREB1 expression, SIN3A ChIP was enriched on the *FGFR4* and *HRAS* promoters and KDM1A ChIP was enriched on the *HRAS* and *MAP2K2* promoters (Fig. 6e). Cells with high RREB1 expression had decreased H3K4me2 on all three promoters compared to control cells (Fig. 6e), consistent with KDM1A as a histone demethylase for H3K4me1/2[40].

In murine erythroleukemia cells, Sin3a binding has been mapped to the *Fgfr4*, *Hras* and *Map2k2* promoters (Fig. 6f). We performed ChIP in *Rreb1*+/− and WT MEFs and hearts to determine how *Rreb1* haploinsufficiency affected occupancy of murine Sin3a and Kdm1a at these promoters. *Rreb1*+/− MEFs had decreased Sin3a on *Fgfr4*, *Hras* and *Map2k2* promoters and decreased Kdm1a on the *Hras* and *Map2k2* promoters compared to WT cells (Fig. 6g). Decreased Sin3a and Kdm1a occupancy was observed on the *Fgfr4*, *Hras* and *Map2k2* promoters in *Rreb1*+/− hearts compared to WT (Fig. 6h). Therefore, MAPK pathway genes are regulated by Rreb1-Sin3a-Kdm1a in murine fibroblasts and cardiac cells.

## Discussion

Our data demonstrate that RREB1 forms a transcriptional repressive complex together with SIN3A and KDM1A which normally leads to transcriptional inactivation of MAPK signaling components. Single allele loss of *RREB1* leads to functional loss of lysine demethylase activity at RREB1 target promoters and hence increased H3K4m2/3 marks leading to enhanced transcription and MAPK pathway activation. We provide genetic and biochemical evidence that dysregulation of *Fgfr4*, *Hras* and *Map2k2* in the heart of *Rreb1* heterozygous mice is deleterious leading to a cardiomyopathy that is reminiscent of the phenotype observed in Noonan-like RASopathies. Therefore, we propose that *RREB1* is a new RASopathy gene that triggers MAPK pathway activation through the loss of epigenetic repressive marks on critical proteins involved in MAPK pathway signaling.

Genetic disorders caused by haploinsufficiency of transcriptional machinery and epigenetic regulators are emerging as drivers of developmental syndromes[41]. We find that *RREB1* haploinsufficiency resembles a RASopathy in both overlap of clinical features and sensitization of RAS-MAPK signaling observed in multiple cell types. In distinction to the known RASopathies, which are single gene disorders, the 6p25.1p24.3 microdeletion syndrome arises via transcriptional dysregulation of multiple RREB1 target genes including *FGFR4*, *HRAS* and *MAP2K2* that result in sensitization of the MAPK pathway to activation. The germline gain-of-function mutations in *HRAS* and *MAP2K2* associated with Costello syndrome and CFC syndrome, respectively, argue that the increased transcription of RREB1 target genes *HRAS* and *MAP2K2* observed with single gene loss of *RREB1* account for the overlapping phenotypes observed in patients with these syndromes and in the *Rreb*+/− hemizygous in mice.

*Rreb*+/− hemizygous in mice are phenotypically similar to the Noonan mouse model observed with gain-of-function mutation in *Ptpn11*[25]. These characteristics include smaller stature, cranial facial dysmorphism and cardiac abnormalities. Mouse models of other RASopathies have been developed[42]

including introduction of a germline G12V mutation in the endogenous *Hras* locus which phenocopied some abnormalities observed in patients with Costello syndrome, including facial dysmorphia and cardiomyopathies[43]. The Costello mice displayed systemic hypertension, vascular remodeling, and fibrosis in the heart which was age dependent and a consequence of abnormal up regulation of the renin–Ang II system[43]. We have observed an age dependent mechanism of cardiac dysfunction in *Rreb*+/− mice. Future studies will address additional phenotypes such as vascular remodeling and the potential alteration of the renin-Ang II system. A mouse model of CFC syndrome with gain-of-function *Braf* mutation leads to craniofacial malformations, congenital heart defects, musculoskeletal abnormalities and growth delay[44]. A mouse model of CFC syndrome harboring a *Map2k2* mutation is yet to be reported. However, we postulate that the similarities in phenotypes between available RASopathy mouse models and the *Rreb1*+/− mice highlight the deleterious effect of overactive RAS-MAPK signaling on organismal development.

Our results show that SIN3A and KDM1A are components of a RREB1 repressor complex and each component contributes to sensitization of MAPK signaling. Haploinsufficiency of *SIN3A* and *KDM1A* mutations lead to similar syndromes characterized by the clinical features observed with *RREB1* haploinsufficiency including developmental delay and craniofacial abnormalities[34,45]. Microdeletion of 15q24 encompassing the *SIN3A* gene and point mutations in *SIN3A* share striking features that include facial dysmorphism, short stature and mild intellectual disability and link SIN3A loss of function to 15q24 microdeletion syndrome[34]. De novo mutations in the *KDM1A* allele have been identified in three individuals who share similar clinical features including facial features, global developmental delay and hypotonia[45]. We propose that RREB1, SIN3A and KDM1A are part of a common genetic pathway that defines a new spectrum of RASopathy-like disorders linked to the genetic perturbation of components of this newly defined silencing complex.

Deletion of the terminal end of chromosome 6 is a clinically recognized syndrome called 6pter-p24 deletion syndrome characterized by developmental delay, cardiac abnormalities and craniofacial dysmorphism thought to be caused by haploinsufficiency of the transcription factor *FOXC1*[46,47]. Terminal deletion of 6p with breakpoints in 6p25.3 represents the vast majority of cases[18]. These syndromes are recognized as distinct from branchiooculofacila syndrome (BOFS) caused by haploinsufficiency or mutation of the transcription factor AP-2α encoded by *TFAP2A* on the centromeric end of 6p24.3[48,49]. Clinical cases with interstitial microdeletion of chromosome 6 in the 6p25.1p24.3 region not involving *FOXC1* or *TFAP2A* display an overlap of clinical features with 6pter-p24 deletion syndrome hinting at a distinct etiology[14–18]. Interestingly, FOXC1 and TFAP2A were both found in proximity to RREB1 using BioID suggesting some target genes may be co-regulated by these transcription factors.

In conclusion, we have provided mechanistic evidence that *RREB1* haploinsufficiency is the pathogenic cause underlying the clinical phenotypes observed in the 6p25.1p24.3 microdeletion syndromes. The clinical features of 6p25.1p24.3 microdeletion cases as well as the physiological and biochemical consequences of RREB1 deficiency described in our study define a novel mechanism for sensitization of MAPK signaling and the development of a Noonan-like RASopathy disorder. *RREB1* haploinsufficiency thus represents a new category of RASopathy-like syndromes arising through transcriptional overexpression of MAPK pathway genes as a result of deregulated epigenetic control.

## Methods

**Ethics.** Informed consent for publication of the clinical details and photographs was provided by the parents of the proband in this report. The authors affirm that the parents of the human research participant provided informed consent for publication of the image in Fig. 1a. Primary human lymphocytes were obtained from proband and parents following informed consent and approval of the Human Ethics Board of the Hospital for Sick Children (Toronto, ON, Canada).

**Antibodies and reagents.** The source of antibodies used in the study were as follows: AbCam H3K9ac (ab4441), H3K27Ac (ab4729). Cell Signaling Technologies AKT (9272), p-AKT (4056, 4058), CTBP1 (8684), ERK (9102), p-ERK (9101), FGFR4 (8562), GAPDH (2118), H3K4me3 (9751), H3K4me2 (9725), H3K36me3 (4909), H3K9me2/3 (5327), HA-tag (3724), HDAC1 (5356), HDAC2 (57156), Histone H3 (14269), JUN (9165), LSD1 (2139), MEK (4694), p-MEK (9154), pan-RAS (3965), RBBP7 (9067), RCOR1 (14567), SIN3A (8056), Stretavidin-HRP (3999). All CST antibodies were used at 1:1000 in 5% milk/PBST buffer. Santa Cruz c-MYC (sc-4084), HRAS (sc-34), KRAS (sc-30), NRAS (sc-519), TUBULIN (sc-69969). Santa Cruz antibodies were used at 1:500 except for TUBULIN at 1:3000. Sigma Aldrich FLAG-M2 (F1804, 1:1000), FGFR4 (HPA-028251, 1:500), RREB1 (HPA-001756, 1:500). The source of growth factors and mitogens were as follows: Cell Signaling Technologies EGF (8916), FGF1 (5234), FGF2 (61972), TGF-beta (8915), TNF-alpha (8902). Origene FGF8 (TP723101). Prospec HGF (cyt-244), FGF16 (cyt-939). Mek2 was a gift from Dustin Maly (Addgene plasmid # 40776; http://n2t.net/addgene:40776; RRID:Addgene_40776).

**Bimolecular fluorescence complementation (BiFC) assay.** The constructs pCMV-HA-VN173-Tubulin and pCMV-HA-VC155-Tubulin were obtained from Dr J. DeLuca (Colorado State University). *RREB1* was cloned into pCMV-HA-VC155-Tubulin, *SIN3A*, *KDM1A* and *HDAC1* were cloned into pCMV-HA-VN173-Tubulin using XhoI and EagI sites and NEBuilder HiFi DNA Assembly cloning kit following manufacturers' protocol. For BiFC analysis, 50 ng of pCMV-HA-VC155-RREB1 was transfected into HEK293 cells seeded onto glass-bottom 24-well plate. Following 24 h of transfection, the cells were subsequently transfected with 50 ng of empty vector, pCMV-HA-VN173-SIN3A or pCMV-HA-VN173-HDAC1. pCAG-CFP was used at 20 ng per well as control. Transfection was performed using LipoD293 reagent. Confocal imaging was performed with an Olympus IX81 inverted microscope using 60x/1.4 PlanApo oil-immersion objective and FluoView software. The excitation wavelengths used were 405 nm and 488 nm for CFP and Venus respectively.

**BioID sample preparation.** RREB1 was cloned into pcDNA-BirA plasmid using standard molecular biology and published protocols. HEK293-TREx flp-in cells (Invitrogen/Thermofisher, Waltham, MA) were co-transfected with pOG44 and pcDNA5 FRT/TO FLAG-BirA-RREB1 using the X-tremeGENE9 DNA transfection reagent (Sigma-Aldrich). Stable lines were selected in DMEM/10% FBS supplemented with 200 µg/ml hygromycin B. Cells were grown to 70% confluency on $5 \times 150$ mm2 dishes and incubated for 24 h in complete media supplemented with 1 µg/ml tetracycline (Sigma) and 50 µM biotin (BioShop). Cells were washed twice with PBS, scrapped from the dish, pelleted by centrifugation ($250 \times g$ for 5 min) and snap frozen. For mass spectrometry experiments, pellets were lysed in 10 ml of RIPA buffer (50 mM Tris-HCl pH 7.5, 150 mM NaCl, 1 mM EDTA, 1 mM EGTA, 1% Triton X-100, 0.1% SDS, 1:500 protease inhibitor (Sigma), 250U Turbonuclease (Accelagen) at 4 °C for 1 h and sonicated (30 s, at 35% power, Sonic Dismembrator 500; Fisher Scientific) to disrupt aggregates. Following centrifugation ($35,000 \times g$ for 30 min), supernatants were incubated with 30 µl of pre-equilibrated Streptavidin-Sepharose beads (GE) at 4 °C for 3 h. Beads were collected by centrifugation (250xg, 2 min), washed with 50 mM ammonium bicarbonate pH 8.2 (six times), and treated with TPCK-trypsin (Promega) for 16 h at 37 °C. The tryptic peptide containing supernatants were collected, lyophilized, suspended in 0.1% formic acid and an aliquot (1/6th the sample) analyzed by mass spectrometry.

**Cell lines and cell culture.** HEK293 were obtained from ATCC. EBV-transformed lymphocytes were created from mouth swabs at the Centre for Applied Genomics (The Hospital for Sick Children, Toronto, ON, Canada). Mouse embryonic fibroblasts (MEFs) were established from E13.5 embryos minced, filtered and cultured in DMEM supplemented with 10% fetal bovine serum. All cultures were maintained in a 5% $CO_2$ environment at 37 °C. EBV-LCL and HEK293T cells were culture in DMEM (Life Technologies Inc.) supplemented with 10% fetal bovine serum (HyClone). pLKO.1 lentiviral vectors expressing shRNAs targeting RREB1 or a nonspecific sequence (shCon) were cotransfected into HEK293T cells with pPAX2 and pVSVG (Addgene) using the X-tremeGENE 9 DNA Transfection Reagent (Roche). The virus was collected 48 hours after transfection. HEK293T cells were infected with lentivirus and stable lines were selected in DMEM/10% FBS/200 µg/ml neomycin and considered stable after 6 days growth in selection. HEK293T-REx cells (Invitrogen/Thermofisher) were co-transfected with pOG44 and pcDNA5 FRT/TO FLAG-BirA-RREB1 using the LipoD293 transfection reagent (SignaGen, Rockville, MD). Stable lines were selected in DMEM/10% FBS/200 µg/ml hygromycin B. Cellular proliferation assay were performed with

$5.0 \times 10^3$ cells plated in 96-well plate ($n = 8$) grown for 24–72 h and treated with Alamar Blue (Thermofisher) according to the manufacturer's protocol.

Colony assays were conducted with HEK293 cells expressing stable shRNA control or shRNA-RREB1 (5000 cells/ml). Briefly, a suspension of 0.35% agarose in DMEM (4.5 mg/ml glucose) supplemented with 10% FBS in the absence of antibiotic was layered onto a 0.5% agarose/DMEM base layer. Cells were permitted to grow for 7–10 days and plates were photographed. Cellular proliferation was measured using Essen IncuCyte ZOOM system, briefly $5.0 \times 10^3$ cells were plated in a 96-well plate ($n = 8$) and cell confluency monitored over the time course. Transient transfections with siRNA were performed using 1–5 nM siRNA (SiGenome-Dharmacon) and RNAiMAX reagent (TherrmoFisher) according to the manufacture's protocol for 72 h.

**Chromatin immunoprecipitation.** ChIP was performed in HEK293-TREx-FLAG-BirA-RREB1 cells or *Rreb+/−* or WT MEFs using SimpleChIP Enzymatic Chromatin IP kit with magnetic beads (Cell Signaling #9003) following the manufacturers' protocol. Efficacy of ChIP is largely determined by the availability of ChIP grade antibodies which are not available for RREB1. Therefore, we used a tetracycline (Tet) inducible flag-tagged RREB1 in the human cell line HEK293 and the flag moiety for immunoprecipitation. All samples for ChIP were conducted on 3 independent experiments in triplicate. Primers for ChIP were designed using Primer3 software (Supplementary Table 1). For ChIP in murine hearts, hearts were excised and perfused with cold PBS-2mM EDTA until pale. Heart tissue was minced and incubated in DMEM, crosslinked with 2% formaldehyde for 20 min with shaking. Cross-linking was quenched with glycine. Cells were pelleted (280g for 10 min) washed with cold PBS and suspended in DMEM with 0.5 mg/ml Collagenase D (Sigma Aldrich) for 50 min. Following digest, cells were washed and suspended into Buffer B from the SimpleChIP Enzymatic Chromatin IP kit, sonicated to break up remaining tissue pieces and digested with 0.05% MN as per the SimpleChIP kit.

**ChIP-seq analysis.** Raw reads were aligned to hg38 using bwa (version 0.7.15) (PMID: 22569178). The resulted sam files are converted to bam with samtools (version 1.8) (PMID: 21245279). MACS2 (version 2.1.1.20160309) (PMID: 18798982) was used to call peak on the bam files. bedGraph files containing signal per million reads produced from MACS2 was converted to bigwig files with ucsc tool kit (315). Selected publicly-available ChIP-seq datasets (GSM803530, GSE91830, GSM803525 and GSE91601) were obtained from the Gene Expression Omnibus (GEO). The results were parsed for genes of interest, the data subsetted and converted to bedGraph format using tools available from the UCSC Genome Browser. Data was imported into R using established packages. Processed ENCODE tracks were filtered for significant peaks based on matched $p$-value tracks downloaded from the same GEO series for each dataset using $p < 0.05$, and annotated using ChIPseeker (v1.12.1).

**Echocardiography.** For echocardiography measurements, mice were anesthetized (2% isoflurane, 98% oxygen) and body temperature maintained at 37 °C. Echocardiography was performed using a 15-MHz linear ultrasound transducer (Vivid7; GE). LV end-diastolic diameter (LVEDD) and LV end-systolic diameter (LVESD) mmode measurements were made from short-axis views at the level of the papillary muscle. LVEDD was measured at the time of the apparent maximal LV diastolic dimension, whereas LVESD was measured at the time of the most systolic excursion of the posterior wall. LV fractional shortening (FS) was calculated as follows: $FS = (LVEDD − LVESD)/LVEDD \times 100\%$.

**Gene expression analysis.** Total RNA was isolated from cells with TRIZOL (Invitrogen) according to the manufacturers' protocol. cDNAs were made using the QuantiTect kit (Qiagen). QPCR was performed using an ABI Step One-Plus System with LUNA Universal master mix (NEB).

**Glycerol gradient column.** Nuclear extract harvested from HEK293T cells transiently transfected with VC-RREB1 (according to protocol described above) was layered on top of a continuous glycerol gradient (10–60%) in Tris buffer containing NaCl. Following ultracentrifugation at 135,000 g for 18 h at 4 °C, fractions were collected, boiled with sample buffer and run on SDS-PAGE for subsequent western blot analysis.

**Immunohistochemistry.** Hearts were dissected from mice and immediately fixed in formalin for 24 h and then placed in 70% ethanol. Cardiac sections from wild type and *Rreb1+/−* hearts were fixed on microscope slides side by side to facilitate direct comparisons of stained tissue. Standard protocols provided by the STTARR (Spatio-temporal Targeting and Amplification of Radiation Response) facility at University Health Network (UHN) were used for IHC.

**MAPK activation assays.** Cells were plated into 12-well plate at 70–80% confluence. Cells were serum starved in DMEM base media for 8 h and then stimulated with FBS (final concentration 10%), or other factors (final concentration 50 ng/ml). EBV-suspension cells were pelleted to remove growth media, washed with PBS, and

serum starved in DMEM base media. Following starvation, $5.0 \times 10^5$ cells were collected in Eppendorf tubes and stimulated with FBS. Cells were pelleted by centrifugation ($280 \times g$ for 10 s). All signaling reactions were quenched immediately after time points with 250 μl 2X sample buffer (100 mM Tris-HCl pH 6.8, 4% (w/v) SDS, 0.2% (w/v) bromophenol blue, 200 mM DTT). Reactions were analyzed by SDS-PAGE/western blot using standard protocols.

**Mice**. C57BL/6J-Rreb1 mice were made at The Centre for Phenogenomics (The Hospital for Sick Children, Toronto, ON, Canada). All mice received environmental enrichment, animal rooms are maintained at 20–24 °C, 40–65% humidity, and 12 h light/dark cycle. Cas9 endonuclease-mediated cleave was used in C57BL/6 J zygotes obtained from the Canadian Mouse Mutant Repository. Cas9 RNA guided recognition sites as follows (1): CCAGCTGACTACATGGCAAG, (2): CTC GGCTGCAGGAAGTACAC, (3): GAAAACTCGTAGTGGCACAG, (4): CGTTA CAACAAAGCACCCTT. All animal studies were approved by the Animal Research Council of the University Health Network (Toronto, Ontario, Canada). All experiments were performed on male mice.

**Promoter assays**. The promoter sequences were amplified by PCR with from human gDNA (Roche) and using Q5 High-fidelity 2x Master Mix (New England Biolabs). Full length PCR products were gel purified (Agarose 0.5% w/v), digested with Fermentas FAST digest reagents (Thermo Scientific) and cloned into the pGL3-Basic vector (Promega) utilizing the NheI/Bgl-II sites. Ligation reactions were conducted with Rapid DNA ligation kit (Roche). The Dual-Luciferase Reporter Assay System (Promega) was used for promoter activation assays. Briefly, $2.0 \times 10^5$ cells were transfected with 100 ng of pGL3-promoter reporter construct and 4 ng of phRL-SV40 (Promega) using Lipofectamine-2000 (Invitrogen) according to the manufacturer's protocol. 18 h post transfection, cells were lysed and assayed for firefly and renilla luciferase activity reading on a Glo-Max dual injector luminometer (Promega). Each measurement was made on 3 distinct samples repeated in triplicate. Error bars for Luciferase activity represent standard deviations.

**RAS activation assays**. Serum starved cells were stimulated with 10% FBS (final concentration) for 1–12 min to activate RAS. Cells were washed with cold PBS, lysed with TX100 buffer (25 mM TRIS pH 7.5, 100 mM NaCl, 5 mM EDTA, 1% Triton) supplemented with protease and phosphatase inhibitors (Roche) and incubated for 1 h at 4 °C with rotation. An aliquot was removed for analysis of total RAS in the whole cell lysate and the remainder incubated with 10ug GST-tagged Raf-binding domain (RBD, gift from Ikura lab). Complexes were immunoprecipitated with magnetic glutathione-agarose beads (Sigma Aldrich). IPs were washed with TX100 buffer and RAS-GTP eluted by boiling in 2X sample buffer.

**RNA-seq analysis**. Transcripts were quantified using STAR (v2.4.2a) and the UCSC genome browser (Feb 2009 assembly, hg38) as the reference and Gencode (v25) for annotation. Differential analysis of quantified read counts from across the samples was facilitated by the DESeq2 (v1.16.1). Transcripts with zero reads mapped across all samples were filtered out prior to downstream analysis. Read counts were collapsed to gene level and transcripts with the highest reads mapped were kept. Lowly-expressed genes were filtered out by applying a minimum sum of at least 10 reads mapped in total per gene across the six samples. Fold changes were generated from the filtered count data matrix, modeled as a function of condition (CON vs. sh-*RREB1*), and p-values were further adjusted for multiple testing using a false discovery rate (FDR) of 1%. Significant hits were defined as genes with an FDR-adjusted p-value of at least 0.01 and an absolute log2-fold change greater than 1.

**Statistical data analysis**. Statistical analysis was performed using the R statistical environment (v3.4.1). Determination of significance for QPCR data was done using Student's t-test, assuming equal variance, and p-values were calculated based on two-tailed test. Functional gene-annotation enrichment analysis, functional annotation, KEGG pathway mapping analysis of CHIP-seq and RNA-seq was performed using DAVID (https://david.ncifcrf.gov/home). All error bars represent standard deviations calculated with Excel. Box plots indicate the interquartile range (IQR) of the data and the central line shows the median calculated using Displayr (https://www.displayr.com/).

**Western blot**. Standard protocols were followed. Proteins were resolved by SDS-PAGE on 8–10% gels or 4–20% wedge gels (Sigma Aldrich). All gels were transferred to PVDF membranes (EMD Millipore). Membranes were blocked in 5% nonfat dried milk in PBS plus 0.1% Tween-20 (PBST). To analyze the nuclear compartment, cells were lysed in hypotonic buffer (10 mM TRIS pH 7.4, 10 mM NaCl, 10 mM EDTA, 0.5% Triton) supplemented with protease and phosphatase inhibitors. Nuclear fraction was pelleted by centrifugation and suspended in 2X sample buffer. Blots were exposed on Microchemi-4.2 (DNR Bio-imaging Systems) and quantified with BIO-RAD Quantity One. All uncropped blots are provided in Supplementary Figs. 7 and 8.

**X-ray imaging**. X-rays were performed using the Faxitron UltraFocusDXA machine (Faxitron Bioptics, LLC, Tucson, Arizona, USA) and the Automatic Exposure Control software. High-resolution X-ray images of skulls were acquired at approximately 25 kV for five seconds and 2× magnification.

**Reporting summary**. Further information on research design is available in the Nature Research Reporting Summary linked to this article.

## Data availability

All relevant data supporting the key findings of this study are available within the article and its Supplementary Information files or from the corresponding author upon reasonable request. The RNA-seq and ChIP-seq data discussed in this publication have been deposited in NCBI's Gene Expression Omnibus and are accessible through GEO Series accession number GSE151417 for RNA-seq and GSE146902 for ChIP-seq. Source data are provided with this paper.

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

## Acknowledgements

The authors thank the patient's family for their participation and feedback. We thank the many colleagues who offered helpful comments and discussion. The work was supported by a CIHR foundation scheme Grant No. 410005207.

## Author contributions

Conceptualization: O.A.K., P.K., R.R. Data generation: O.A.K., M.S., E.C., H.E.B., R.X.S., N.L., K.D., S.C., K.B., E.L., J.S., Y.M., J.C., A.M.S., J.L. Formal analysis: O.A.K., M.S., H.E.B., B.R., F.B., R.R. Methodology: O.A.K. Supervision: H.H., B.R., F.B., R.R. Figures: O.A.K. Writing – first draft: O.A.K. Writing – final review & editing: O.A.K., R.R.

## Competing interests

The authors declare no competing interests.
