## [Peer Review File · Nature Communications]

Peer Review Information

Manuscript title: Haploinsufficiency of RREB1 causes a Noonan-like RASopathy via epigenetic reprogramming of RAS-MAPK pathway genes.

Corresponding author name(s): Dr. Oliver Kent and Prof. Robert Rottapel

Editorial notes:

none

Reviewer comments & decisions:

Reviewer comments, first version:

Reviewer #1 (Remarks to the Author):

This paper identifies a role of RREB1 in the pathogenesis of a Noonan-like syndrome. Starting from a clinical observation the authors identify RREB1 as a protein that represses the expression of several ERK pathway genes by recruiting demethylases and deacetylases to the promoters of these genes. They also generate a RREB1 heterozygous knockout mouse that re-capitulates the clinical phenotype. Overall, this is a very exciting study that reports a novel way for the genesis of rasopathies, identifies the gene causing it and does an excellent job in tracing the pathophysiological mechanisms. They also use a variety of methods including knock-out mouse technology, transcriptomics, ChIPseq, and interaction proteomics to support their findings. This paper is highly suitable for Nature Communications and highly recommendable for publication. I only have some minor points that the authors may wish to address.

Fig. 1F. The hyperactivation of the ERK pathway is difficult to see on these blots. A longer exposure of the P1/P2 pERK blots probably would look very similar to the Boy pERK blot. These samples should be all on the same blot to be comparable. The ratio between pERK and ERK shown in Fig. S1d does not allow a comparison between the three individuals unless all samples were on the same blot.

Figs. 3C. according to the methods ChIP experiments were performed with tetracycline induced RREB1. This is not clear from the text and should be stated. As with all overexpression experiments there is concern that the overexpressed protein may have 'spill over' functions, in this case bind to promoters where it normally would not bind to. The authors should address that, at a minimum by including a Western blot comparing RREB1 expression plus / minus tetracycline.

Fig. 3E. It would be interesting to see whether RREB1 overexpression has the reverse effect to shRNA. This would be re-assuring in terms of excluding possible off-target effects of the two shRNAs.

Reviewer #2 (Remarks to the Author):

In their manuscript entitled "Haploinsufficiency of RREB1 causes Noonan-like syndrome via epigenetic reprogramming of RAS-MAPK pathway genes" Oliver Kent and colleagues start from the description of a patient with clinical features consistent with a Noonan-spectrum disorder and identify RREB1 as the causative gene. The authors provide in vitro and in vivo evidence that haploinsufficiency of RREB1 triggers MAPK pathway activation via loss of its repressive function. Furthermore, the authors show that SIN3A, KDM1 and RREB1 are components of a repressive complex and suggest they form part of a genetic pathway that could cause RASopathy-like disorders via epigenetic alterations.

The manuscript is well written, the experiments are comprehensive and include all necessary controls. Data are represented in an understandable way and conclusions drawn are well grounded on the data. The finding that loss of epigenetic repression of components of the RAS-RAF-MEK-ERK pathway can cause a Noonan-like disorder is novel and interesting.

Below, I have listed a few points of criticism that should be addressed by the authors:

Major point:

- The authors mention that two other genes, LY86 & SSR1 are also located in the minimally deleted region. The authors should comment on their potential contribution to the phenotype. What is the actual evidence that these genes contribute (or not) to the observed phenotype?

Minor points:

- Some figure panels are too small and one can hardly see the relevant information, this is especially true for items displaying pathological or histological information (e.g. Fig. 2e, 2f, 2i), but also for the photo from the colony assay (Fig. 4c) and for the ChIP-seq tracks (e.g. Fig. 3c, 3e, 5d, etc.).
- The y-axis of the ChIP-seq tracks should indicate units and the data range shown.
- The figure legends in general should contain sufficient information in order to unambiguously understand what exactly is displayed. This is not the case for almost all main and supplementary figures.
- Figure 4d should be accompanied by a quantification of the bands as I cannot recapitulate the findings that the author state in the text regarding this figure.
- The y-axis in Fig 6f does not cover the entire data range resulting in some of the peaks being cut. This should be changed.
- The authors should support their findings by stating the results of statistical testing for all experiments where this applies (i.e. Fig. 1e, 2d, 2e, 2f, 2g, 5d, 5e, 5f, 6d, 6e, 6g, 6h).

Reviewer #3 (Remarks to the Author):

Kent et al. identified a patient with an interstitial 6p25.1q24.3 microdeletion who exhibited short stature, mild intellectual disability, and widely spaced eyes. Of 11 genes deleted in 6p25.1q24.3, RREB1, a transcription factor activated downstream of RAS signaling, is the common deleted gene in patients with multiple interstitial microdeletion of 6p. The authors demonstrated that Rreb1^{+/-} mice showed craniofacial abnormalities and left ventricular hypertrophy with cardiac cell hypertrophy. The ChIP and RNAseq analysis showed that RREB1 is a negative regulator of FGF-HRAS-MAPK signaling. They showed Rreb1 recruits Sin3a and Kdm1a to control H3K4 methylation at MAPK pathway gene promoters.

This is an interesting report that identify novel target genes or epigenetic mechanisms of RREB1 gene. However, I am wondering if clinical manifestations in patients with interstitial deletion of 6p25.1p24.3 could be relevant to those in patients with Noonan syndrome. In addition, the pathogenesis could be different from RASopathies.

1. It is difficult to conclude that the patient in this study could have overlapping features with those in Noonan syndrome. Did the patient have relative macrocephaly, dawnslanting palpebral fissures, pigmented skin, pectus deformity, or EEG abnormalities which are typical in patients with Noonan syndrome? The authors could address the detailed clinical manifestations and could consult to experienced dysmorphologists for the clinical diagnosis of RASopathy patients.
2. As the authors cited, several patients with interstitial deletion of 6p25.1p24.3 have been reported. Patients with 6p25.1p24.3 deletion had congenital anomaly syndrome with intellectual disability, including craniofacial dysmorphism, cardiac anomaly or hearing loss. Have the authors re-evaluated the clinical manifestations of the patients with 6p25.1p24.3 deletion reported before? Were the patients diagnosed as having Noonan syndrome? Alternatively, have any patients with mutations in RREB1 reported?
3. In fig. S3m, protein expression of FGFR4, HRAS, MEK1/2 was increased in heart tissues in Rreb1^{+/-} mice compared with those in WT. HRAS germline mutations cause Costello syndrome and MAP2K2 germline mutations cause CFC syndrome, not Noonan syndrome.
4. ERK was activated in heart tissues and MEFs from Rreb1^{+/-} mice (Fig.2i,j). Model mice with HRAS G12V and G12S had hypertrophic cardiomyopathy, cardiac fibrosis or cardiac valve abnormalities. In addition, western blotting showed that ERK was not activated in all tissues from H-RAS G12V mice (Schumacher et al. J Clin Invest. 2008). The authors could address the difference between Rreb1^{+/-} mice and RASopathy model mice.
5. In Figure 4f, the authors showed that RAF-binding activity of pan-RAS or HRAS was increased in RREB1- deficient cells. It is not clear why GTP-bound HRAS was increased? Is FGFR4 expression increased in HEK293 cells expressing shRREB1? Increased expression of FGFR4 directly activated HRAS? In figure S3h, protein expression of HRAS was increased in HEK293 cells expressing shRNA targeting RREB1. The authors could address the mechanisms how HRAS itself is activated.
6. In Figure 4d, treatment of EGF, FGFR16 or HGF showed enhanced activation of ERK in sh-RREB1 cells. How did these ligands activate ERK in sh-RREB1 cells? Upregulation of FGFR4 are associated with the EGF or HGF signaling pathways?
7. In Figure 5b and 5c, robust activation of K4me3, K4me2, K36me3, and K9 acetyl were observed in lysates from HEK293 cells expressing sh RREB1 and Rreb1^{+/-} MEFs. Is only KDM1A itself associated to the increased H3Kme2 and K4me3?

Author rebuttal, first version:

Reviewer #1. I only have some minor points that the authors may wish to address:

Fig. 1F. The hyperactivation of the ERK pathway is difficult to see on these blots. A longer exposure of the P1/P2 pERK blots probably would look very similar to the Boy pERK blot. These samples should be all on the same blot to be comparable. The ratio between pERK and ERK shown in Fig. S1d does not allow a comparison between the three individuals unless all samples were on the same blot.

The reviewer is absolutely correct, the blots are difficult to interpret correctly unless these experiments were performed at the same time, on the same lysates and under the same conditions of HRP development. We performed the westerns on uncut membranes to ensure the levels of p-ERK and ERK were developed under the same conditions (blocking, washes, secondary antibody) and the same exposure time so as to enable direct comparison of the bands. During the organization of the figures, we changed the order of the parental (P1/P2) and boy samples by cropping the original images to match other elements in the figure. We thought this was clearly stated in the figure legend and thank the reviewer for pointing out we did not state this explicitly. We have provided the uncut original view of the membranes in Supplemental Figure 1 to give the reader a direct comparison of the signals on the original blots. In addition, we have clearly indicated in the figure legend of Fig.1 that these westerns were originally developed on the same membranes but separated following exposure for the figure.

Figs. 3C. according to the methods ChIP experiments were performed with tetracycline induced RREB1. This is not clear from the text and should be stated. As with all overexpression experiments there is concern that the overexpressed protein may have 'spill over' functions, in this case bind to promoters where it normally would not bind to. The authors should address that, at a minimum by including a Western blot comparing RREB1 expression plus / minus tetracycline.

As the reviewer pointed out, we used a tetracycline induced RREB1 for ChIP-seq and ChIP experiments. We apologize for not describing this experiment with sufficient clarity in the main text. As such we have added the sentence,

"Since ChIP grade antibodies are not available for RREB1, we created a tetracycline inducible flag-tagged RREB1 cell line for ChIP experiments. Treatment of cells with tetracycline activated RREB1 mRNA expression 10-fold over untreated cells with RREB1 protein over-expression detected at physiologic levels (Supplementary Fig.3a,b)."

We have provided the QPCR analysis of *RREB1* mRNA expression and western blots of protein levels in the supplementary figures to demonstrate the RREB1 induction by tetracycline treatment results in physiologic levels of RREB1. Our goal in using this "tunable" approach was to emulate physiologic levels of tagged RREB1 rather than analyzing cells with excessively high

levels of RREB1 expressed from the transgene. The two experiments have been added to Supplementary Fig.3a,b respectively. We believe we are not analyzing a “spill-over” effect of RREB1 expression as suggested by the reviewer since RREB1 protein levels using this induction method are relatively low and within the physiologic range.

Comment to the Editor: Due to the amount of data contained in Supplementary Figure 3, we moved the MAPK genes identified by RNA-seq (previously SupFig.3e) to the main figure 3 (now Fig.3c). Further, the validation of RREB1 target genes by western blot in the heart lysates (previously SupFig.3m) has been moved to the main figure (now Fig.3j). Relocating these two figure panels created the necessary additional space in the supplementary figure 3 for the new supplement data described above and moved important but previously buried data to the main figure 3.

Fig. 3E. It would be interesting to see whether RREB1 overexpression has the reverse effect to shRNA. This would be re-assuring in terms of excluding possible off-target effects of the two shRNAs.

We agree that examination of promoter activation in cells with enforced RREB1 over expression is a great experiment. However, due to the COVID-19 pandemic we have been unable to perform any experiments as our laboratory, the Princess Margaret Cancer Centre and Research Institute and University of Toronto have been closed. As an alternative, we previously examined the endogenous expression of RREB1 target gene mRNAs using RT-QPCR of the three main genes described in the paper in RREB1 over expressing HEK293 cells. We find that *FGFR4*, *HRAS*, and *MAP2K2* mRNAs are repressed in cells with enforced RREB1 expression in complete agreement with other experiments in the paper. We have added these data to Fig.3. We added the following sentence to the manuscript to describe the data:

“Consistent with a mechanism of RREB1 repression of target genes, HEK293 cells with the tetracycline inducible flag-tagged RREB1 had repressed levels of *FGFR4*, *HRAS* and *MAP2K2* mRNAs when cells were treated with tetracycline to induce RREB1 expression (Fig.3g).”

These data are directly comparable to promoter activation and provide evidence of endogenous gene repression by RREB1. These data reinforce our observations seen when RREB1 was knocked down using shRNAs and address the concern of the reviewer regarding the potential off target effects of the shRNA constructs.

Reviewer #2. I have listed a few points of criticism that should be addressed by the authors:

Major point: The authors mention that two other genes, *LY86* & *SSR1* are also located in the minimally deleted region. The authors should comment on their potential contribution to the phenotype. What is the actual evidence that these genes contribute (or not) to the observed phenotype?

We apologise for this misunderstanding as the error is completely a consequence of our ambiguous text. The LY86 and SSR1 genes are the nearest upstream and downstream neighbours of RREB1 respectively, which we included simply as positive controls for genes contained within the microdeletion region. We validated the expression of these two genes, in addition to RREB1, in the EBV-LCL cells derived from the boy and his parents as mRNA positive controls for the deleted region. Our wording may have contributed to the misapprehension that these genes contribute to observed phenotypes. We have therefore rewritten the sentence, “Two other genes contained within the deleted region, SSR1 and LY86, were similarly decreased in LCLs from the proband compared to parents” to “We validated two additional genes contained within the deleted region, SSR1 and LY86, and found they were similarly decreased in LCLs from the proband compared to parents.”

We present data that strongly argue that these genes do not contribute to the sensitization of MAPK signaling observed with RREB1 haploinsufficiency. First, the minimal region of overlap as determined by analysis of genomic data for microdeletion cases reveals that only the RREB1 gene is commonly observed as indicated in Fig1. Second, LY86 is not expressed in HEK293 cells under normal conditions and therefore cannot contribute to altered RAS signaling we observe when RREB is knocked down. While SSR1 is expressed in HEK293 cells, knockdown of SSR1 with two independent shRNAs did not alter MAPK pathway activation. These data are now included in the supplemental Fig4 (panels b,c,d).

We added the following paragraph to describe this data: “Multiple genes are commonly lost in the 6p25.1p24.3 microdeletion cases; therefore we wanted to demonstrate that decreased expression of RREB1 rather than decreased expression of neighboring genes caused sensitization of MAPK signaling. We examined the expression of the nearest neighbor genes to RREB1 in HEK293 cells and found they only express *SSR1* in addition to *RREB1* (Supplementary Fig.4b). Targeting *SSR1* using two independent shRNAs had no effect on p-MEK and p-ERK compared to sh-control cells (Supplementary Fig.4c,d) confirming a RREB1 specific role for MAPK pathway regulation.”

Some figure panels are too small and one can hardly see the relevant information, this is especially true for items displaying pathological or histological information (e.g. Fig. 2e, 2f, 2i), but also for the photo from the colony assay (Fig. 4c) and for the ChIP-seq tracks (e.g. Fig. 3c, 3e, 5d, etc.).

We have enlarged hearts images Fig2e and Fig.2i. We have provided an enlarged inset of the H&E staining in Fig.2f to clearly display the cells and combined with the quantification (Fig.2g) emphasizes the major finding of hypertrophic CM in the *Rreb1*^{+/-} deficient hearts. We have enlarged the images of the colony assay in Fig4c and we have enlarged the ChIP-seq traces in Fig.3c, 3e, 5d, and 6f as requested.

The y-axis of the ChIP-seq tracks should indicate units and the data range shown.

Thank you for pointing this out. We have added the y-axis units and data range to the ChIP-seq tracks in Figures 3d, 3f, 5d, 6e, and 6f as requested. We also added the y-axis data to

Supplementary Fig.5e. Furthermore, we have added the ENCODE source of these data since we realized during editing they were omitted in the first version. This source information has been added to the figure legends.

The figure legends in general should contain sufficient information in order to unambiguously understand what exactly is displayed. This is not the case for almost all main and supplementary figures.

We apologize for the brevity of the figure legends. We have expanded all main and supplementary figure legends in attempt to clearly describe what is being displayed.

Figure 4d should be accompanied by a quantification of the bands as I cannot recapitulate the findings that the author state in the text regarding this figure.

We have added the quantification of p-ERK signals at the 5 and 21 minute time points to the figure. We have added minor changes in descriptive words used in the text to correlate and accurately describe the quantification of the western blots.

The y-axis in Fig 6f does not cover the entire data range resulting in some of the peaks being cut. This should be changed.

Thank you for pointing this out. We have adjusted the y-axis levels so that the peaks are not cut off. We have replaced the ChIP-seq plots in the original figure with the adjusted plots. We have also included the y-axis data range which was mentioned in the comment above as missing.

The authors should support their findings by stating the results of statistical testing for all experiments where this applies (i.e. Fig. 1e, 2d, 2e, 2f, 2g, 5d, 5e, 5f, 6d, 6e, 6g, 6h).

We agree that our results should be explicitly supported by statistical reference where applicable. Therefore, we have evaluated the RREB1 expression (Fig1e) and provided a statistical measure in the figure which we reference in the text. We have added reference to other significant findings (Fig.2d, 2g, 2h, SupFig.2g) as pointed out by the reviewer to the text.

In addition to the experiments specifically referred to by the reviewer, we have also provided a statistical measure of ChIP experiments (5d,5e,5f,6e,6g,6h) and provided a statistical measure of mRNA QPCR experiments (Fig3g,3h,3i,6d,SupFig3h,3j,3m) as to be all inclusive within the paper. The statistical measures have been added to the figures and figure legends.

Comment to the Editor: We realized during the analysis of ChIP data to determine the statistical significance and during preparation of the source data file that Fig.5f and 6h MAP2K2 ChIP data were presented with the wrong axis displayed in the figure. We have fixed this error. We also realized that Fig.6g FGFR4 data was from a CUT and RUN experiment and not ChIP as we wanted. We have changed the data in that panel to the ChIP data. The overall interpretation does not change, however the magnitude of the expression is different from the previous version.

Reviewer #3.

1. It is difficult to conclude that the patient in this study could have overlapping features with those in Noonan syndrome. Did the patient have relative macrocephaly, dawnslanting palpebral fissures, pigmented skin, pectus deformity, or EEG abnormalities which are typical in patients with Noonan syndrome? The authors could address the detailed clinical manifestations and could consult to experienced dysmorphologists for the clinical diagnosis of RASopathy patients.

We agree that the term Noonan syndrome not be appropriate to describe the phenotype of our proband child and therefore have used the term “Noonan-like” instead since the child manifests some of the physical findings seen in Noonan and he has evidence of activated RAS MAPK signaling. Clinical geneticists agree that there is a great variability in expression and the phenotype of Noonan syndrome which can attenuate with increasing age. According to Allanson and Roberts (Noonan Syndrome, GeneReviews), the key features of the Noonan syndrome facies include low-set, posteriorly rotated ears (present in the proband); vivid blue or blue-green irises (not present); and eyes that are often wide-spaced (present in proband), downslanted (present in proband), and with epicanthal folds and fullness or droopiness of the upper eyelids (not present). The child has no cafe au lait macules, pectus deformity or pulmonary valve stenosis. Congenital heart disease occurs in 50%-80% of individuals.

A scoring system for the diagnosis of Noonan syndrome has been proposed (Orphanet J Rare Dis. 2007; 2: 4) but it is not universally used by clinical geneticists. According to these criteria, the proband has suggestive facial dysmorphology, short stature and mental retardation. There are a number of conditions with phenotypes strikingly similar to Noonan syndrome caused by variants in the RAS signaling pathway. This is why we have chosen to label the patient described in this study as a ‘Noonan-like syndrome’.

2. As the authors cited, several patients with interstitial deletion of 6p25.1p24.3 have been reported. Patients with 6p25.1p24.3 deletion had congenital anomaly syndrome with intellectual disability, including craniofacial dysmorphism, cardiac anomaly or hearing loss. Have the authors re-evaluated the clinical manifestations of the patients with 6p25.1p24.3 deletion reported before? Were the patients diagnosed as having Noonan syndrome? Alternatively, have any patients with mutations in RREB1 reported?

We have only evaluated the patients with 6p25.1p24.3 microdeletion reported in the literature using the information provided within the respective references. We have presented this phenotypic information in the paper as thoroughly as we are able from information available in these literature sources. None of the patients in any of these case studies, as far as we know, were diagnosed with Noonan or Noonan-like syndromes. However, in the paper by Qi et al. (Qi et al. BMC Medical Genomics 2015), they state: “The RAS/RAF/MEK/ERK signal transduction pathway is known to be associated with Noonan syndrome (OMIM #163950), Costello syndrome (OMIM #218040) and Cardio-Facial-Cutaneous syndrome (OMIM# 115150). These syndromes share some of the phenotypic features of the 6p25.1p24.3 deletions, such as

cardiac abnormalities, craniofacial dysmorphism and hemangioma. Therefore, we speculate that deletion of the RREB1 gene may underlie some, if not all, of these phenotypes through the RAS/RAF signal pathway.” Based on this statement, we suspect they also were considering the similarities of the 6p25.1p24.3 syndrome to the RASopathy syndromes. Patients with germline mutations in the RREB1 gene have been reported in the ClinVar database; however, the consequences of mutation on phenotype manifestation are not reported.

3. In fig. S3m, protein expression of FGFR4, HRAS, MEK1/2 was increased in heart tissues in Rreb1+/- mice compared with those in WT. HRAS germline mutations cause Costello syndrome and MAP2K2 germline mutations cause CFC syndrome, not Noonan syndrome.

As the Reviewer pointed out, mutations in genes encoding the MAPK pathway genes HRAS and MAP2K2 lead to Costello and CFC syndromes respectively. We agree with the Reviewer that assigning a generalized tag to the syndrome observed with *RREB1* haploinsufficiency is difficult and likely not accurate when one considers the genetic cause rather than the observed clinical phenotypes. We have struggled with assigning the proper nomenclature of *RREB1* haploinsufficiency syndrome. We have conferred closely with the clinical geneticists at the Hospital for Sick Children about this issue in addition to our review of the literature. We have considered referring to the *RREB1* haploinsufficiency syndrome as Noonan, Noonan-like or RASopathy. We feel, given the issue has raised by this Reviewer and also considering our responses to questions 1 and 2 above, that we should use the more general term “RASopathy” which is used collectively to describe Noonan, Costello, CFC and other similar syndromes. We have changed all mention of “Noonan-like” in the text to “RASopathy”. However, we like the title of the paper to read “Noonan-like RASopathy” instead of “Noonan-like syndrome” providing the reader with some reference as to the meaning of RASopathy which may not be entirely clear to all readers.

4. ERK was activated in heart tissues and MEFs from Rreb1+/- mice (Fig.2i,j). Model mice with HRAS G12V and G12S had hypertrophic cardiomyopathy, cardiac fibrosis or cardiac valve abnormalities. In addition, western blotting showed that ERK was not activated in all tissues from H-RAS G12V mice (Schumacher et al. J Clin Invest. 2008). The authors could address the difference between Rreb1+/- mice and RASopathy model mice.

We agree with the reviewers comments that different models of RASopathy mice driven by different RAS family members and activating alleles demonstrate different cardiac phenotypes. We have included the following in the text to the discussion which properly provides context of the cardiac phenotypes observed in the *Rreb1* heterozygous mice with the phenotypes of other RASopathy model mice:

“*Rreb*+/- hemizygous in mice are phenotypically similar to the Noonan mouse model observed with gain-of-function mutation in *Ptpn11* (25). These characteristics include smaller stature, cranial facial dysmorphism and cardiac abnormalities. Mouse models of other RASopathies have been developed (42) including introduction of a germline G12V mutation in the endogenous *Hras* locus which phenocopied some abnormalities observed in patients with

Costello syndrome, including facial dysmorphism and cardiomyopathies (43). The Costello mice displayed systemic hypertension, vascular remodelling, and fibrosis in the heart which was age dependent and a consequence of abnormal up regulation of the renin–Ang II system (43). We have observed an age dependent mechanism of cardiac dysfunction in *Rreb*^{+/-} mice. Future studies will address additional phenotypes such as vascular remodelling and the alteration of the renin-Ang II system. A mouse model of CFC syndrome with gain-of-function *Braf* mutation leads to craniofacial malformations, congenital heart defects, musculoskeletal abnormalities and growth delay (44). A mouse model of CRC syndrome harbouring a *Map2k2* mutation is yet to be reported. However, we postulate that the similarities in phenotypes between available RASopathy mouse models and the *Rreb1*^{+/-} mice highlight the deleterious effect of overactive RAS-MAPK signaling on organismal development.”

We have also added three additional references (#42,43,44 and including Schumacher et al. J Clin Invest. 2008 mentioned by the Reviewer) that we make reference to in the above paragraph to the reference list in the paper.

5. In Figure 4f, the authors showed that RAF-binding activity of pan-RAS or HRAS was increased in RREB1- deficient cells. It is not clear why GTP-bound HRAS was increased? Is FGFR4 expression increased in HEK293 cells expressing shRREB1? Increased expression of FGFR4 directly activated HRAS? In figure S3h, protein expression of HRAS was increased in HEK293 cells expressing shRNA targeting RREB1. The authors could address the mechanisms how HRAS itself is activated.

The increased HRAS activation in *RREB1* deficient cells is plausibly the combined effects of increased *HRAS* expression (gene dosage effect) together with increased *FGFR4* expression and activity. Consistent with this statement, we observed more *HRAS* mRNAs and protein levels in multiple cell types tested with decreased *RREB1* expression. Because we observe increased GTP-bound HRAS in *RREB1* deficient cells in response to FBS, EGF, FGFs we conclude that the MAPK pathway signaling is sensitized by increased *HRAS* expression to a number of growth factor stimulae. We have added the following sentence to the text to make this clearer: “This result is consistent with increased gene dosage of *HRAS* mRNA and protein levels resulting in increased HRAS activation.” This sentence is followed by: “Therefore, we tested the importance of *HRAS* expression in sensitizing MAPK pathway activation” in which we then described how *HRAS* knockdown in *RREB1* deficient cells abrogated the sensitization phenotype, thus tying together these two ideas.

6. In Figure 4d, treatment of EGF, FGFR16 or HGF showed enhanced activation of ERK in sh-RREB1 cells. How did these ligands activate ERK in sh-RREB1 cells? Upregulation of FGFR4 are associated with the EGF or HGF signaling pathways?

We performed the experiment in Fig.4d to ascertain the mechanism of activation of ERK signaling downstream of specific receptors. We hypothesized based on mRNA expression analysis (Fig3) that the *FGFR4* receptor would likely be involved due to its high expression in multiple cell types with *RREB1* haploinsufficiency. As mentioned in the comment above, we

suspect a gene dosage effect likely potentiates increased MAPK pathway activation downstream of *FGFR4*, *HRAS* and *MAP2K2* genes collectively. Although we do see increased ERK activity following stimulation with FGF16 (specific for *FGFR4*), we also see increased activation of ERK when cells are stimulated with EGF and HGF likely through different receptors and not involving *FGFR4*. The activation of p-ERK by these ligands would likely involve EGF stimulation of an EGF-receptor and HGF activation of c-MET (or other) receptor followed by downstream activation of *HRAS*. Activation of any or all of these receptor systems would result in the increased activation of *HRAS* as described in the text and in our response to the question above. These data suggest that *HRAS* and *MAP2K2* levels are a signaling bottleneck downstream of multiple growth factors alleviated by increased expression in *RREB1* haploinsufficient or knockdown cells.

7. In Figure 5b and 5c, robust activation of K4me3, K4me2, K36me3, and K9 acetyl were observed in lysates from HEK293 cells expressing sh RREB1 and Rreb1+/- MEFs. Is only KDM1A itself associated to the increased H3Kme2 and K4me3?

We feel there may be some confusion of the Reviewer to the mechanism of action with KDM1A. We have shown that loss or decreased *RREB1* and by association loss or decreased *KDM1A* via a complex recruited by *RREB1* to promoters, leads to an increase of *H3K4me2* and *K4me3* in all cell types tested (*HEK293*, *MEFs*, and heart cells). Our work shows that *H3K4* methylation at a *RAS* response element is regulated by the *RREB1-KDM1A-SIN3A* complex. The activity of *KDM1A* on promoters controls the initiation of transcription via regulation of *H3K4* methylation. The enzymatic activity of *KDM1A* controls the removal of a methyl mark from *H3K4me2* substrates. We find that when *RREB1* is decreased due to shRNA knockdown or haploinsufficiency, increased levels of *H3K4me2* are found at target promoters which we believe results from decreased recruitment of *KDM1A* to these promoters. We also found that *H3K4me3* levels are increased on these promoters. The demethylase(s) for *H3K4me3* are currently unknown

Reviewer comments, second version:

Reviewer #1 (Remarks to the Author):

The authors have addressed all my comments and I find the manuscript is acceptable for publication

Reviewer #2 (Remarks to the Author):

The present revised version of the manuscript reads very well and the authors have appropriately addressed all the points previously raised by the reviewers.

Reviewer #3 (Remarks to the Author):

The authors have answered all of my questions and the paper has been significantly improved. The

manuscript contains novel findings on target genes or epigenetic mechanism of *RREB1*. Further identification of patients with germline *RREB1* mutations will clarify the effect of *RREB1* mutations on human development.

Author rebuttal, second version:

Reviewer comments, third version:

Author rebuttal, third version: